# Computational analysis of peripheral blood smears detects disease-associated cytomorphologies

José Guilherme de Almeida [1,8], Emma Gudgin[2], Martin Besser[2], William G. Dunn [2], Jonathan Cooper[3], Torsten Haferlach [4], George S. Vassiliou [3,5,6] ✉ & Moritz Gerstung [1,7] ✉

Many hematological diseases are characterized by altered abundance and morphology of blood cells and their progenitors. Myelodysplastic syndromes (MDS), for example, are a group of blood cancers characterised by cytopenias, dysplasia of hematopoietic cells and blast expansion. Examination of peripheral blood slides (PBS) in MDS often reveals changes such as abnormal granulocyte lobulation or granularity and altered red blood cell (RBC) morphology; however, some of these features are shared with conditions such as haematinic deficiency anemias. Definitive diagnosis of MDS requires expert cytomorphology analysis of bone marrow smears and complementary information such as blood counts, karyotype and molecular genetics testing. Here, we present Haemorasis, a computational method that detects and characterizes white blood cells (WBC) and RBC in PBS. Applied to over 300 individuals with different conditions (*SF3B1*-mutant and *SF3B1*-wildtype MDS, megaloblastic anemia, and iron deficiency anemia), Haemorasis detected over half a million WBC and millions of RBC and characterized their morphology. These large sets of cell morphologies can be used in diagnosis and disease subtyping, while identifying novel associations between computational morphotypes and disease. We find that hypolobulated neutrophils and large RBC are characteristic of *SF3B1*-mutant MDS. Additionally, while prevalent in both iron deficiency and megaloblastic anemia, hyperlobulated neutrophils are larger in the latter. By integrating cytomorphological features using machine learning, Haemorasis was able to distinguish *SF3B1*-mutant MDS from other MDS using cytomorphology and blood counts alone, with high predictive performance. We validate our findings externally, showing that they generalize to other centers and scanners. Collectively, our work reveals the potential for the large-scale incorporation of automated cytomorphology into routine diagnostic workflows.

The diagnosis of hematological malignancies relies on expert cytomorphological examination of blood, bone marrow and/or other tissue biopsies, together with molecular analyses that aid subclassification and prognosis[1]. For example, anemias, characterized by reduced hemoglobin concentration (Hb) and altered red blood cell (RBC) numbers, can be both a disease and a feature of other conditions such as myelodysplastic syndromes (MDS), a heterogeneous group of myeloid neoplasms that can progress to acute myeloid leukemia

(AML)[2–4]. For this reason, the diagnosis and further subtyping of MDS requires the detection of cytopenias, changes to white blood cell (WBC) and RBC maturation blood cell through cytomorphologic analysis of bone marrow (BM) and peripheral blood slides (PBS), cyto- and histochemistry, karyotyping and immunophenotyping[4–8].

An accurate diagnosis of MDS and other hematological malignancies is essential to guide treatment: while megaloblastic anemia (MA), which can be confused with MDS[9–11], is generally treated with dietary changes or supplements[12], the treatment of MDS generally involves chemotherapeutic agents, blood/platelet transfusions and hypomethylating agents[13,14] and depends on risk stratification which considers blood counts, BM cytomorphology and cytogenetics[15]. Furthermore, MDS prognosis can also benefit from molecular genetics, used to define clinically-relevant MDS subtypes such as *SF3B1*-mutant MDS that is associated with improved survival times[16,17]. It should be noted that MDS cases with splicing factor mutations such as *SF3B1*-mutant MDS account for over 50% of all cases[18,19], constituting an important MDS subtype.

While abnormalities such as an increased prevalence of hypolobulated granulocytes, abnormal granularity in neutrophils or abnormal RBC are common in MDS[8,20–22], peripheral blood cell morphology is generally insufficient for MDS diagnosis. This is compounded by challenges in the assessment of subtle cytomorphological alterations and heterogeneity across any given PBS leading to inter-observer variation. While diagnoses stemming from the analysis of a PBS (requiring the analysis of hundreds of cells) typically show high concordance, the classification and characterization of individual WBC is more challenging[23,24]. Additionally, the evidence on whether trained experts can distinguish specific cell types is conflicting[25,26], and a study looking specifically at cell type classification concordance among 28 morphologists showed that experts agreed on only 60% of all classified cells[27]. This creates challenges in identifying relevant cytomorphology-disease associations. Computational methods, which have shown promise in the characterization and prognostication of MDS and AML using bone marrow slides[28–30] and identification of abnormal leukocytes[31], can help address some of these problems.

Here we present Haemorasis, a machine-learning protocol that automatically detects and characterizes blood cells in PBS, and apply it to a cohort of individuals with MDS or anemia demonstrating its use in predicting diseases and deriving novel "morphotypes", associations between cellular morphology and different blood conditions. We show that *SF3B1*-mutant MDS can be distinguished from other MDS using cytomorphology and blood counts alone with high predictive performance, with hypolobulated neutrophils and large RBC being more prevalent in this MDS subtype. Using expert-annotated WBC and RBC, we show that virtual cell types are enriched in commonly recognized WBC and RBC types/abnormalities. Finally, we externally validate our approach, showing that it largely generalizes to different centers and WBS scanners.

## Results

### The MLL cohort captures previously described clinical features of MDS and anemia

The MLL cohort was composed of 203 male and 159 female individuals, with mean age 66.1 (362 individuals in total). Individuals with MDS were older than the remaining MLL cohort, with a bias towards males as previously reported[32]—the chance of having MDS in our cohort increased by 12% every year, with males being more than twice as likely to have MDS ($p = 8 \times 10^{-16}$ and $p = 0.00017$, respectively, for the binomial regression of MDS diagnosis based on age and sex; Fig. 1a, b; Table 1).

Additionally, for the linear regression of WBCC against binary MDS and anemia (vs. Normal), MDS and deficiency anemias were associated with leukopenia (1,200 ($p = 0.04$) and 1,800 ($p = 0.009$) fewer WBC/μL respectively (Fig. 1c). However, this leukopenic tendency in anemias was driven by MA—whereas IDA was indistinguish-

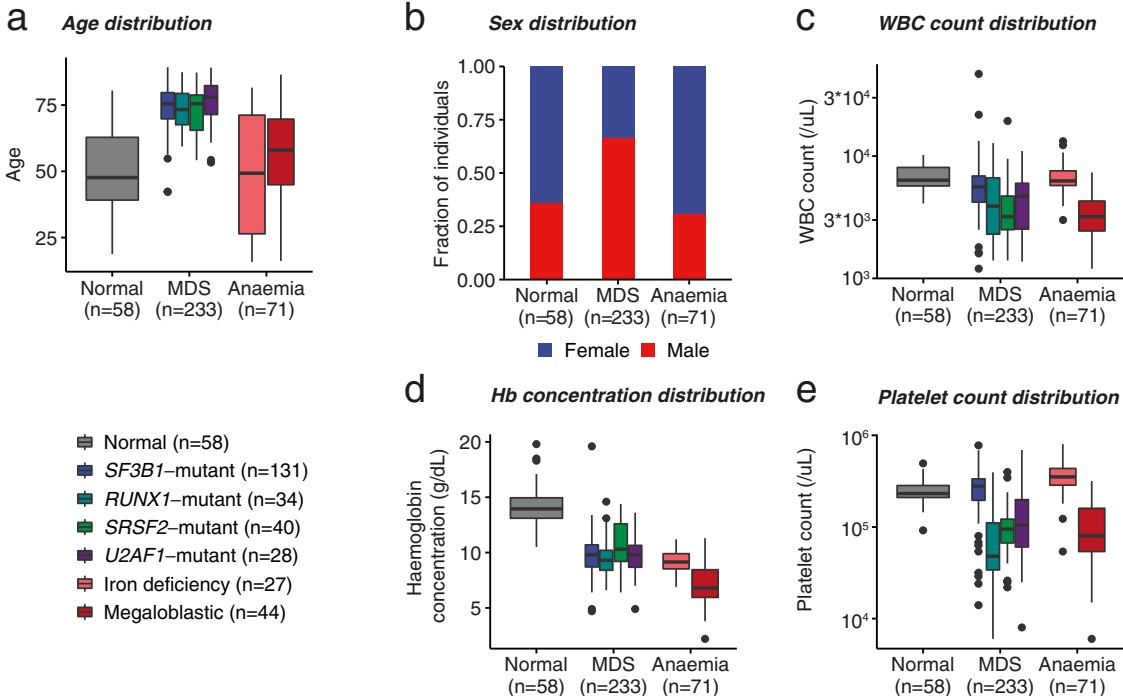

**Fig. 1 | General features of anemias and myelodysplastic syndromes (MDS) in the MLL cohort. a, b**—Age and sex distributions of individuals according to different conditions, respectively. **c**–**e** White blood cell (WBC) counts, hemoglobin concentration and platelet counts, respectively, according to different conditions.

For **a, c, d, e** the boxplots represent the minima and maxima (whiskers), 25% and 75% quantile (upper and lower edges of the box) and the median (the central line of the box), with outliers (defined as points not contained within 1.5 IQR (interquantile range) of the 25% and 75% quantiles) signaled with black dots.

**Table 1 | Statistical comparisons of different features of the MLL cohort**

| Variable | Coefficient | Standard error/difference estimate | p-value | Model | Formula |
|---|---|---|---|---|---|
| Age | 0.11 | 0.01 | $8 \times 10^{-16}$ | Binomial regression | Has MDS ~ age + sex |
| Sex (is male) | 1.20 | 0.32 | $2 \times 10^{-4}$ | | |
| Normal (vs. baseline) | 6750 | 556 | $<2 \times 10^{-16}$ | Linear regression | WBCC (/uL) ~ Condition |
| MDS (vs. normal) | −1272 | 610 | 0.04 | | |
| Anemia (vs. normal) | −1880 | 713 | $9 \times 10^{-3}$ | | |
| Normal (vs. baseline) | 14.29 | 0.30 | $<2 \times 10^{-16}$ | Linear regression | Hb (g/dL) ~ Condition |
| MDS (vs. normal) | −4.35 | 0.32 | $<2 \times 10^{-16}$ | | |
| Anemia (vs. normal) | −6.39 | 0.38 | $<2 \times 10^{-16}$ | | |
| Normal (vs. baseline) | $256 \times 10^3$ | $23 \times 10^3$ | $<2 \times 10^{-16}$ | Linear regression | Plt (/uL) ~ Condition |
| MDS (vs. normal) | $-48 \times 10^3$ | $26 \times 10^3$ | 0.06 | | |
| Anemia (vs. normal) | $-40 \times 10^3$ | $30 \times 10^3$ | 0.2 | | |
| Normal | $256 \times 10^3$ | $[111 \times 10^3, 182 \times 10^3]$ | $3 \times 10^{-12}$ | Two-sided t-test | Plt (/uL) ~ Megaloblastic anemia vs. normal |
| Megaloblastic anemia | $110 \times 10^3$ | | | | |
| *SF3B1*-mutant MDS | $282 \times 10^3$ | $[131 \times 10^3, 200 \times 10^3]$ | $<2 \times 10^{-16}$ | Two-sided t-test | Plt (/uL) ~ *SF3B1*-mutant MDS vs. other MDS |
| Other MDS | $117 \times 10^3$ | | | | |
| *SF3B1*-mutant MDS | $282 \times 10^3$ | $[-62 \times 10^3, 10 \times 10^3]$ | 0.15 | Two-sided t-test | Plt (/uL) ~ *SF3B1*-mutant MDS vs. normal |
| Normal | $256 \times 10^3$ | | | | |
| *SF3B1*-mutant MDS | 9.80 | [−0.83,0.22] | 0.25 | Two-sided t-test | Hb (g/dL) ~ *SF3B1*-mutant MDS vs. other MDS |
| Other MDS | 10.10 | | | | |
| *SF3B1*-mutant MDS | 6253 | [655,2801] | $2 \times 10^{-3}$ | Two-sided t-test | WBCC (/µL) ~ *SF3B1*-mutant MDS vs. other MDS |
| Other MDS | 4525 | | | | |
| *SF3B1*-mutant MDS | 6253 | [−1515,522] | 0.3 | Two-sided t-test | WBCC (/µL) ~ *SF3B1*-mutant MDS vs. normal |
| Normal | 6750 | | | | |

When **Model** is "Binomial regression" or "Linear regression", the **Coefficient** column refers to the coefficient in the linear regression; when the **Model** is "Two-sided t-test", the **Coefficient** column refers to the mean value for each **Variable**.

**Table 2 | Differences between cohorts (MLL vs. CUH2) regarding age and blood counts (*Hb* hemoglobin concentration, *Plt* platelet count, *WBCC* WBC counts) stratified by condition (Control; *IDA* iron deficiency anemia, *MA* megaloblastic anemia, *SF3B1*-mutant - *SF3B1*-mutant MDS; Other - Other MDS subtypes)**

| Condition | | Age (years) | | Hb (g/dL) | | Plt (1000/uL) | | WBCC (1000/uL) | |
|---|---|---|---|---|---|---|---|---|---|
| | | MLL | CUH2 | MLL | CUH2 | MLL | CUH2 | MLL | CUH2 |
| Control | | 33.1* [21,44] | 50.6* [18.8,86.0] | 13.0* [11.5,15.4] | 14.3* [10.5,19.8] | 208* [172,244] | 256* [92,496] | 7.2 [5.4,9.2] | 6.8 [4.1,10.2] |
| Anemia | IDA | 47.3 [4,95] | 48.8 [15.8,81.6] | 7.1* [4.6,9.5] | 9.2* [6.9,11.2] | 382 [191,824] | 376 [54,805] | 7.0 [3.7,11.2] | 6.9 [3.0,13.2] |
| | MA | 69.7* [68,72] | 56.3* [16.2,86.5] | 5.3 [2.7,6.9] | 7.2 [2.2,14.6] | 54 [29,97] | 113 [6,318] | 3.3 [2.5,4.9] | 3.9 [1.2,17.5] |
| MDS | *SF3B1* mutant | 66.2 [52,78] | 73.7 [53.3,89.1] | 10.7 [6.7,14.1] | 10.1 [4.9,14.6] | 135 [35,293] | 115 [6,694] | 4.3 [1.6,18.8] | 4.5 [1.4,19.4] |
| | Other | 71.9* [52,85] | 74.0* [42.3,89.3] | 9.27 [5.0,13.1] | 9.80 [4.7,19.6] | 353 [55,861] | 283 [14,780] | 5.6 [2.5,10.4] | 6.2 [1.2,47] |

Signaled with "*" are the values for which statistically different differences were found according to a two-sided t-test comparing both cohorts (MLL and CUH2). The presented values are the median and, in brackets, the range.

able from controls, MA had approximately 3200 fewer WBC/$\mu L$ than controls ($p = 6 \times 10^{-14}$ for a two-sample t-test) as in previous studies[10,33,34]. Hb was also much lower in MDS and anemias (Fig. 1d)—indeed, the Hb of these individuals was lower than that of normal individuals by 4.34 and 6.38 g/dL, respectively ($p < 2 \times 10^{-16}$ and $p < 2 \times 10^{-16}$, respectively, for the linear regression of Hb against binary MDS and anemia diagnosis indicators). No difference between controls and MDS or anemia cases was observable with regards to platelet counts (Plt), but MA had approximately 146,000 fewer platelets/$\mu L$ than controls ($p = 3 \times 10^{-12}$ for two sample t-test; Fig. 1e) in keeping with previous reports[33].

Finally, *SF3B1*-mutant MDS displayed distinct features compared to *SF3B1*-wt MDS—particularly, WBC and Plt were comparable to those of controls and higher than those found in *SF3B1*-wt MDS ($p = 0.3$ and $p = 0.15$ for two sample t-tests comparing WBC and Plt between *SF3B1*-mutant MDS and controls; $p = 0.002$ and $p < 2 \times 10^{-16}$ for two sample

t-tests comparing WBC and platelet counts, respectively, between *SF3B1*-mutant and *SF3B1*-wt MDS), in keeping with previous reports[16,17].

To validate the disease prediction findings we will report ahead, we also digitized slides for the CUH2 cohort (**Methods**) and compared it with the MLL cohort in terms of age and blood counts. We found statistically significant differences in Hb and Plt in controls ($p = 0.009$ and $p = 0.002$, respectively, for two-sided t-tests comparing between cohorts; Table 2), all of which are a likely consequence of the difference in ages ($p = 4 \times 10^{-7}$). Finally, we also found relatively small but statistically-significant differences between Hb in IDA ($p = 0.001$) and age in MA ($p = 0.0001$) and other MDS subtypes ($p = 0.001$).

**Computational cytomorphology of peripheral blood slides**

We detected cells in PBS using Haemorasis (Fig. 2a). For the first stage of this method, quality control of PBS tiles, we trained a DL model to predict whether specific tiles are of "good" or "poor" quality

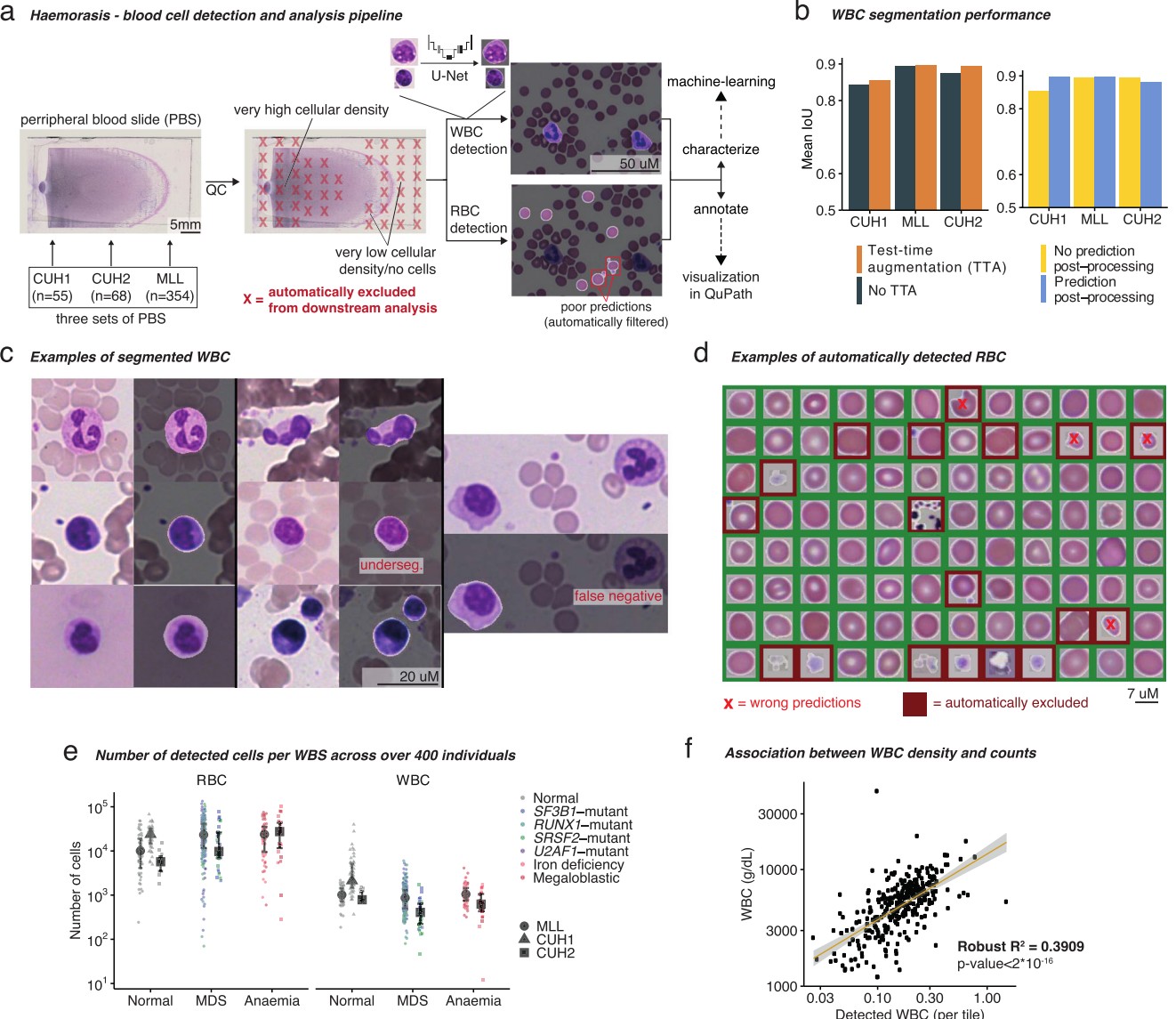

**Fig. 2 | Haemorasis – automated detection and analysis of blood cells in peripheral blood slides (PBS). a** Haemorasis: detection and characterization of blood cells in PBS using computer vision and machine-learning. First, the digitized PBS goes through a deep-learning-based quality control (QC) algorithm that filters out parts of the PBS which are too high or low in cellular density or too blurred. Then, white blood cells (WBC) and red blood cells (RBC) are detected separately–WBC are detected using a U-Net model, a deep-learning algorithm for segmentation, while RBC are detected using simple computer vision methods and filtered using machine-learning. Following this, each individual cell is characterized in terms of shape, texture and color distribution, and annotated for visualization in QuPath[77].

**b** U-Net performance on WBC segmentation. Segmentation post-processing was tested on test-time augmented images. **c** Randomly extracted and representative WBC detection examples and possible errors (underseg. = image under-segmentation error). **d** RBC detection examples and examples of wrongly detected and filtered RBC. Here, all RBC were detected using a simple computer vision protocol, but wrong detections were filtered out using machine-learning (red background). **e** Number of detected blood cells stratified by clinical classification. **f** Association between number of detected WBC using our protocol and WBC counts (two-sided robust $R^2 = 0.39$, $CI_{95\%} = [0.30, 0.49]0$).

(Supplementary Fig. S1a). This (i) reduces the inclusion of non-cellular objects in downstream analyses, thus reducing artifact-associated variation and (ii) limits processing to the clinically-relevant part of the PBS (usually hematologists will consider <20% of the total area; **Supplementary Results**). Next, we detect both WBC and RBC on "good" quality tiles. To detect WBC, we trained a U-Net-based[35] DL model on a dataset of >2800 manually annotated WBC in PBS from CUH1. Extensive data augmentation (random image alterations; Supplementary Table S3) were used to make the model more robust. We validated this model on test sets from CUH1, CUH2 and MLL, with test time augmentation (TTA) improving predictions and prediction post-processing greatly reducing the number of false positive WBC predictions (Fig. 2b, Supplementary Fig. S2a-c). We confirmed the

good performance of the model through visual inspection (Fig. 2b, c, Supplementary Fig. S2d, e) and, while some errors were detected (Fig. 2c), these were small and rare with the model performing well across different cohorts (Supplementary Results; Fig. 2b). RBC were detected using a simple computer vision protocol and predictions were filtered using XGBoost, a fast and scalable machine-learning algorithm[36] (Supplementary Methods; Supplementary Results; Fig. 2d), ensuring that non-RBC objects in PBSs were removed and reducing the rate of false positives from 17.3% (the false positive rate (FPR) in the training dataset) to 1.9% (the product of the validation FPR–11%–of our RBC filtering model FPR and the original FPR in the dataset; in other words, only 1 out of 50 RBC candidates predicted as RBC are false positives).

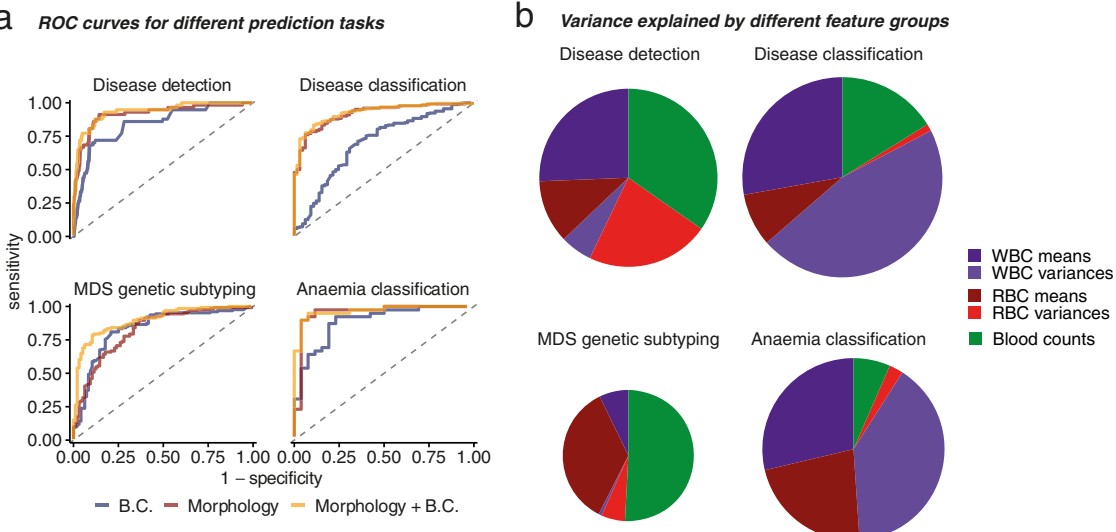

**Fig. 3 | Morphometric moments improve prediction. a** Cross-validated receiver operating characteristic (ROC) curves for the four considered tasks. **b** Feature group contribution to different tasks from 4 models trained on both morphology and blood counts (B.C.). In **a**, lines and bars are colored according to the used datasets for each task. In **b**, circles are scaled according to the total explained variance and coloured according to the feature group.

Across all cohorts, for each PBS we detected an average of 26,000 (range 70 to 133,916) RBC per PBS (a total of 12,042,425 RBC) and around 1,400 (range 12 to 39,862) WBC/PBS (a total of 646,952; Fig. 2e)). The cellular density for the MLL cohort was on average smaller by 44% for RBC/mm² and 10.5% for WBC/mm² compared to CUH (Supplementary Table S4; Supplementary Fig. S3a). Further heterogeneity was observed across conditions, with controls having the highest WBC density and the lowest RBC density (28.9 WBC/mm², 189 RBC/mm²), anemia having the highest RBC density (383 RBC/mm²) and MDS having the lowest WBC density (13.8 WBC/mm²; Supplementary Fig. S3b). Lastly, we also noted that automated blood films produced a higher fraction of good tiles compared to manually prepared slides while controlling for cohort and condition−an additional 5%, highlighting the utility of standardization (Supplementary Fig. S3c).

In line with the findings in Fig. 1c, we extracted on average more cells in controls than in individuals with either MDS or anemia (Supplementary Table S4), although heterogeneity across slides rendered this trend statistically insignificant. Generally, the cellular density of detected WBC in the PBS correlated with WBCCs from automated analysers, validating our detection protocol through an orthogonal approach (robust $R^2 = 0.39$, $CI_{95\%} = [0.30, 0.49]$; Fig. 2f), and demonstarting that we detect a representative number of WBC in PBS. Finally, we characterized all individual cells using morphological features used in other morphometric software programs[37−39] (Supplementary Table S5; Supplementary Fig. S3). For each cell, we quantified its size, shape, color distribution and texture and for WBCs we also characterized their nuclear size and shape (Supplementary Fig. S4). We note here that our method for WBC nuclei segmentation underperforms in conditions of low contrast (where nucleus and cytoplasm are hard to distinguish) or high granularity (particularly for eosinophils and basophils; Supplementary Fig. S5), leading us to focus on cases of high contrast and avoiding conclusions pertaining to eosinophils or basophils.

### Morphological heterogeneity informs disease prediction

We test four distinct tasks to determine whether Haemorasis can be used to meaningfully predict conditions from PBS: (i) disease detection, (ii) disease classification, (iii) MDS genetic subtyping and (iv) anemia classification. Morphometric moments (feature mean and variance across all cells in a PBS) differed across different conditions (Supplementary Results; Supplementary Fig. S6). This qualitative assessment was corroborated by fitting a binomial elastic-net regression model (glmnet)[40] for each task using morphometric moments in addition to WBCC, Hb and Plt. Performance was evaluated using 5-fold cross-validation (the data were split into 5 non-overlapping validation sets while the rest was used for training, leading to less biased models[41]).

Morphometric regression showed high cross-validated predictive performance across all tasks (Fig. 3a, Supplementary Fig. S7a), including an AUC of 89.7% for MDS genetic subtyping (Supplementary Fig. S7a). Additionally, blood counts are highly predictive of *SF3B1*-mutant MDS as indicated in Fig. 1c, e and previous publications[16,17] (Fig. 3, Supplementary Fig. S7a). Notably, morphological feature variance had a significant impact on prediction, revealing that cytomorphological heterogeneity is important for diagnosis (Fig. 3b), as previously suggested for red cell distribution width (RDW)[42]. Finally, the relative importance of different features revealed important trends: for instance, *SF3B1*-mutant MDS was characterized by higher Plt, larger RBC and smaller WBC nuclear area (Supplementary Fig. S7b). However useful, this protocol makes retrieving illustrative examples of blood cells more challenging: larger RBC or more irregular WBC are easily understandable morphometric changes, but changes in morphometric *variance* do not permit satisfactory explanations, making the pictorial demonstration of their importance more elusive.

### Discovering diagnostically relevant morphotypes

At first inspection, the two-dimensional representation of the distribution of cytomorphological characteristics of different conditions revealed an interwoven landscape without immediately recognizable cell clusters (Fig. 4a). However, it becomes apparent that different parts of the cytomorphology space are differentially populated by different conditions. To partition this space and define morphotypes (disease-associated cytomorphological phenotypes), we use a MIL approach that clusters cells based on their cytomorphological characteristics such that the resulting computational morphotypes (CMs) become relevant to the aforementioned diagnostic tasks (Fig. 4b; Supplementary Methods).

We performed Morphotype analysis simultaneously considering the four objectives described earlier and established stable morphotypes consistently found through 5-fold cross-validation (Supplementary Methods). Morphotype analysis performed similarly to morphometric moment prediction when predicting conditions (Fig. 4;

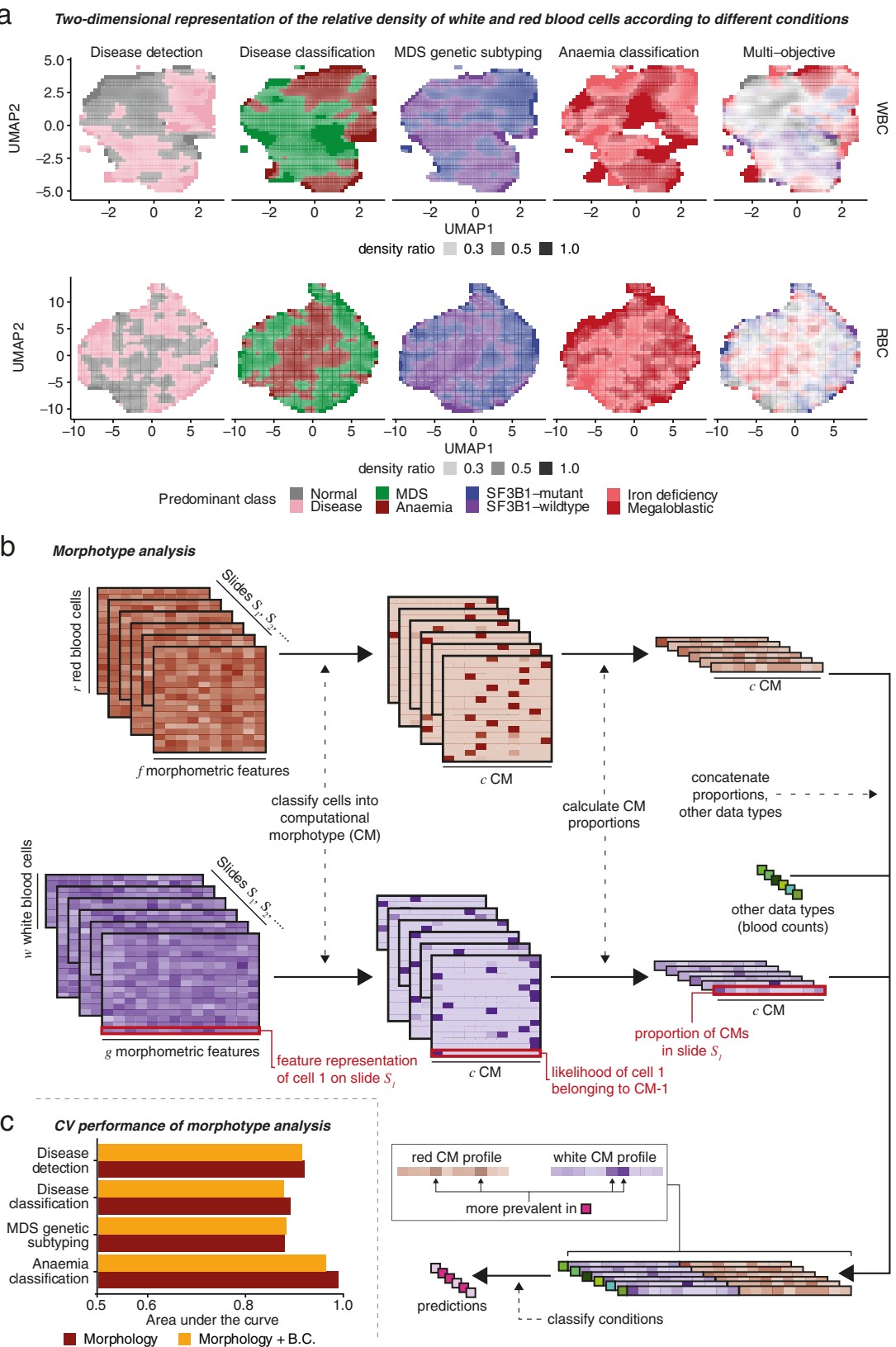

**Fig. 4 | Morphotype analysis. a** Two-dimensional UMAP density ratios of the features of individual WBC (top) and RBC (top). Density ratios are calculated by dividing the local density for the predominant class by the sum of the local density of all classes. **b** Schematic representation of Morphotype analysis. Individual WBC and RBC are clustered into computational morphotypes (CMs) and the proportions of each morphotype on each slide is used to predict different conditions. This allows us to identify which morphotypes are associated with different conditions. **c** Performance of prediction using Morphotype analysis.

**a** *Computational morphotype proportions across multiple tasks using morphotype analysis*

**b** *White computational morphotypes in morphotype analysis*

**c** *Red computational morphotypes in morphotype analysis*

**Fig. 5 | Computational morphotypes across conditions. a** The relevant prevalence of disease-associated computational morphotypes. For this heatmap, the 5 morphotypes with the highest absolute difference in median effect size between conditions are selected for each task, and proportion ratios were calculated as the ratio of the median proportion for each condition. **b** White blood cell morphotypes (WCMs) for different conditions from Morphotype analysis. **c** Red blood cell morphotypes (RCMs) for different conditions from Morphotype analysis. The labeling for WCMs in **b** and the WBC heatmap in **a** is the same, as well as the labeling for RCMs in **c** and the RBC heatmap in **a**.

Supplementary Fig. S8, Supplementary Fig. S9), with the added benefit of producing human-interpretable, disease-associated cytomorphologies. To further demonstrate this, we provide an online visualization tool that allows readers to observe the visual cohesion of different morphotypes (https://josegcpa.github.io/haemorasis-umap; Supplementary Methods). This approach revealed 8 stable WBC morphotypes (denoted WCM 1–8), accounting for 60% of WBCs in normal samples, as well as 12 stable RBC morphotypes (RCM 1–12) comprising 90% of

RBCs (Supplementary Fig. S10). These stable WBC and RBC morphotypes displayed distinct cytomorphological characteristics, while the remaining morphotypes were found to be of variable nature. Among the stable morphotypes, 7 WCMs and 7 RCMs exhibited robust associations with specific clinical conditions (Fig. 5a-c, Supplementary Fig. S11).

Among the stable WBC morphotypes, four mostly consisted of different neutrophil morphologies (WCM-1,2,3 and 4 in Fig. 5a, b),

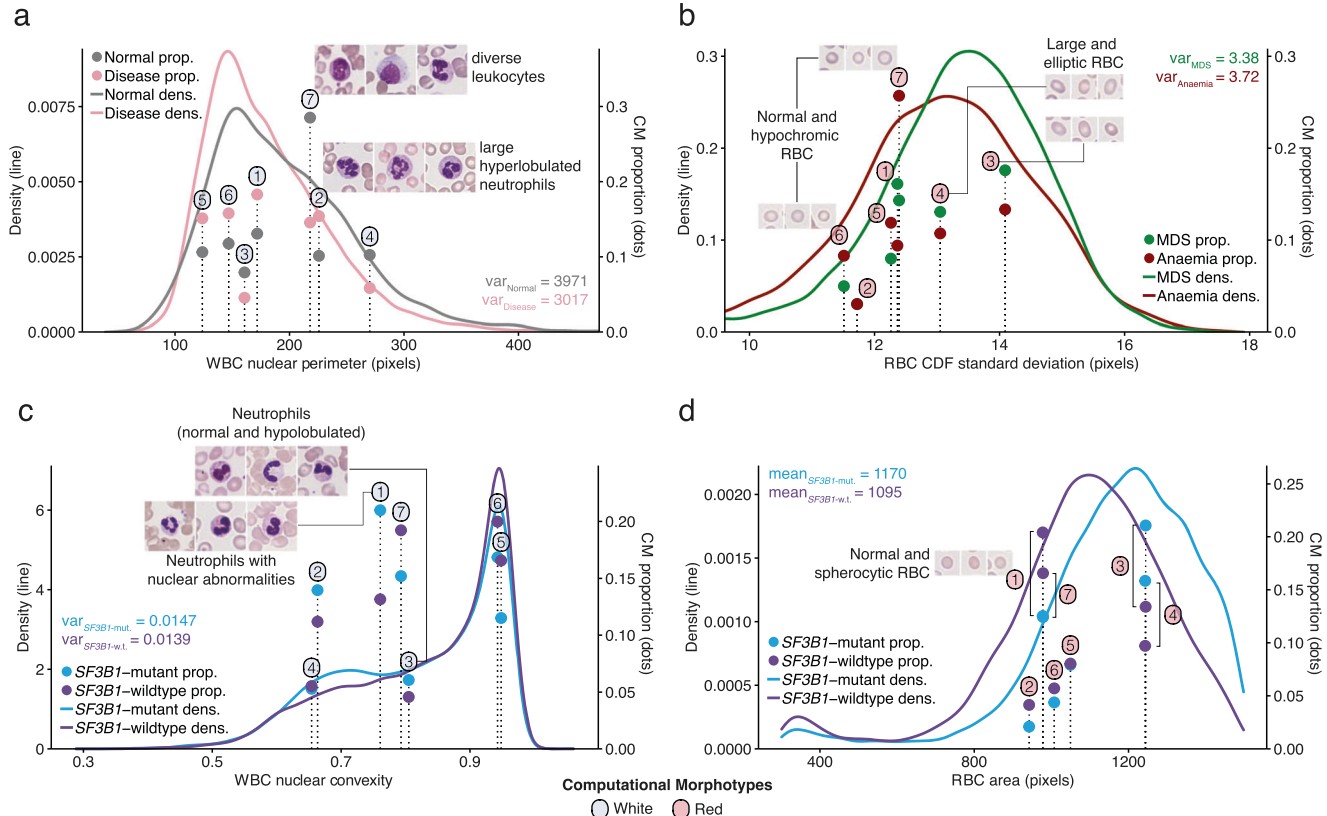

**Fig. 6 | Computational morphotype proportions explain morphometric feature distributions. a–c** Examples of the relationship between condition-specific WCM (left column, **a**, **c**) and RCM (right column, **b**, **d**) proportions and differences in morphometric feature variances for disease detection (WBC nuclear perimeter), disease classification (RBC standard deviation of the centroid distance function (CDF)) and MDS genetic subtyping (WBC nuclear convexity), respectively. **d** Example of the relationship between condition-specific morphotype proportions and differences in RBC area mean for MDS genetic subtyping. Morphometric feature distributions and stable morphotype classifications were derived using a subset of 31,119 RBC and 31,884 WBC from the MLL cohort. The positions of the bars corresponding to different morphotypes are calculated as the mean feature value for each morphotype. Each label corresponds to the nearest bar top, and only morphotypes presented in Fig. 6 are annotated.

highlighting their cytomorphological diversity and diagnostic relevance. WCM-5 contained small lymphocytes, WCM-6 larger lymphocytes and myeloid progenitors, whereas WCM-7 consisted of diverse myeloid cells. We confirmed clinically-relevant cellular phenotypes such as the increased prevalence of abnormal neutrophils in cases of MDS and deficiency anemia (WCM-1 and 2); in MDS, lymphocytes (WCM-5) were less prevalent while immature myeloid cells (WCM-6) are more prevalent as previously suggested[43,44]. Morphotype analysis also identified novel morphotypes—particularly, WCM-3 (normal hypolobulated neutrophils) appeared to be more prevalent in *SF3B1*-mutant MDS, and larger and/or hyperlobulated neutrophils were more prevalent in MA than in IDA (WCM-2 and 4). We confirm these using single-objective Morphotype analysis, where Morphotype analysis models are trained on a single task (Supplementary Fig. S12). Finally, we found that WCM-5 (small lymphocytes) were more prevalent in anemia when compared with MDS.

Stable RBC morphotypes showed more subtle differences. Some morphotypes were relatively more normal—RCM-1 and 2 contained mostly normal or spherocytic RBCs (Fig. 5a, c)—whereas others (RCM-3 and 4) captured larger RBC and elliptocytes. RCM-5 captured relatively small RBC and some poikilocytes, and RCM-6 and 7 captured hypochromic RBCs. We show that RCM-6 and 7 (hypochromic RBCs) were more typical of anemia than MDS as previously reported in IDA[1], with RCM-7 being more prevalent in IDA compared to MA. RCM-3 and 4 (large RBC and elliptocytes) were more prevalent in *SF3B1*-mutant MDS compared with *SF3B1*-wt MDS, while RCM-5 (poikilocytic RBC) were more common in IDA compared to MA.

Notably, morphotype frequency offers more tangible explanations for the associations of morphometric moments with certain diagnoses. If, on average, certain morphotypes are more prevalent in a specific condition, this will manifest as a relation with shifts in the means and/or heterogeneity (variances) of different features (Fig. 6, Supplementary Fig. S13). For example, the variance of the WBC nuclear perimeter, shown to be important for disease detection (Supplementary Fig. S13a), can be explained by the differential frequencies of different WCMs (Fig. 6a): the higher prevalence of WCM-7 drives the increased heterogeneity of this feature in normal individuals. In disease classification, we can further observe how the increased variance of RBC shape irregularity (standard deviation of the centroid distance function) in anemia compared to MDS is partly explained by the elevated prevalence of RCM-5, 6, and 7 (relatively circular, some poikilocytes) and lower prevalence of RCM-3 and 4 (larger and more elliptic; Fig. 6b). Finally, WBC nuclear convexity exhibits a stronger bimodality and therefore greater variance in *SF3B1* mutant MDS cases, driven in part by WCM-1 and 3 (Fig. 6c). Finally, the clear increase in the mean of RBC area in *SF3B1*-mutant MDS, is due to the higher prevalence of RCM-3 and 4 and lower prevalence of RCM-1 and 7 in *SF3B1*-mutant MDS (Fig. 6d).

**Computational cytomorphology validation**
To confirm the nature of the computational morphotypes and their diagnostic associations, we performed: (i) a blinded annotation of cell types by expert clinical hematologists and (ii) a validation of their predictive value in the CUH2 cohort. First, we assessed whether the

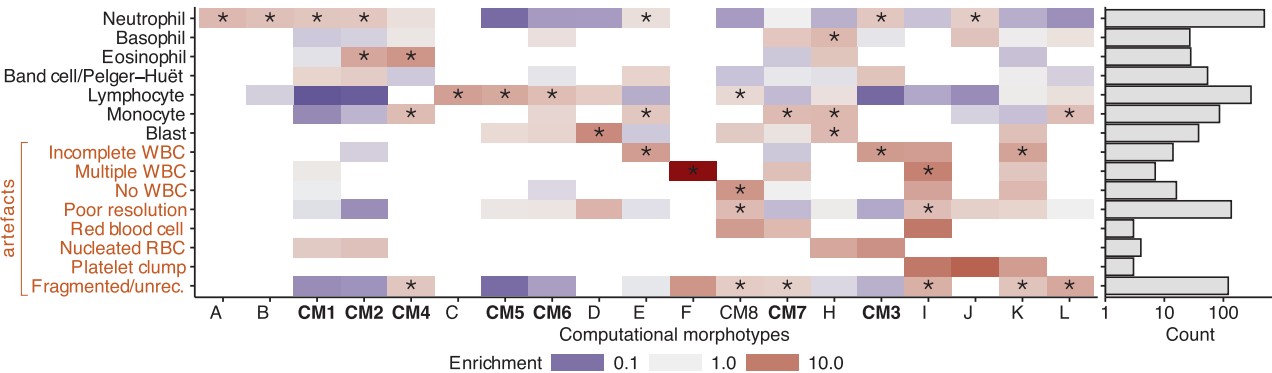

a   *WBC computational morphotype enrichment in expert-annotated WBC types*

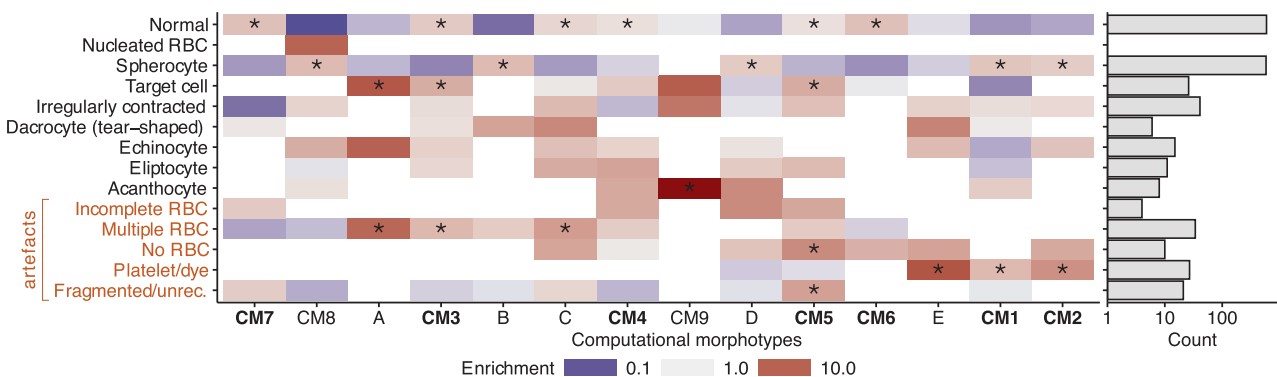

b   *RBC computational morphotype enrichment in expert-annotated RBC types*

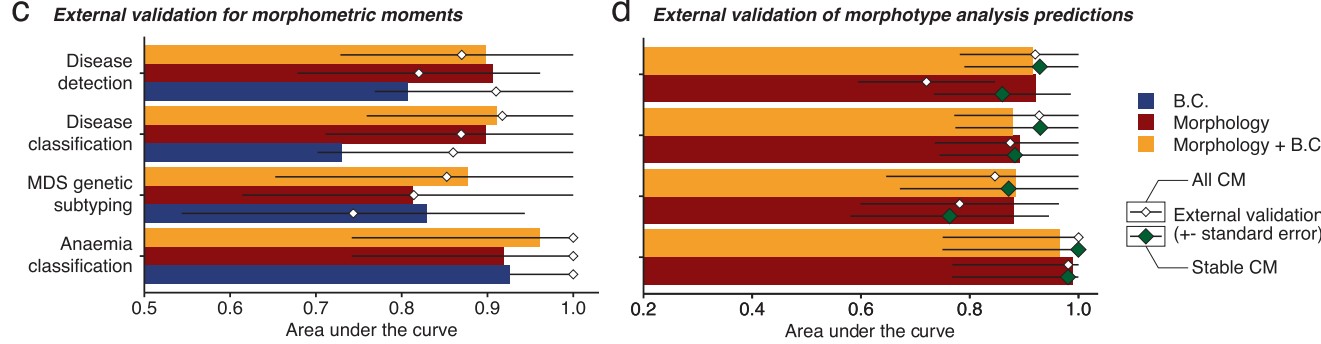

c   *External validation for morphometric moments*

d   *External validation of morphotype analysis predictions*

**Fig. 7 | Expert and external validation of computational cytomorphology.**
**a** Correspondence between WCMs and expert annotated WBC types (left) and number of annotated WBC types. **b** Correspondence between RCMs and expert-annotated RBC types (left) and number of annotated RBC types. **c** External validation for glmnet. **d** External validation performance for the Morphotype analysis with all morphotypes (small white diamonds) and using only stable morphotypes (green diamonds). Enrichment values in **a** and **b** marked with an asterisk "*" are

significant for a chi-squared test. Consensus morphotypes are highlighted in bold and labeled according to Fig. 6 and preceded by "CM", whereas uncertain CM are labeled arbitrarily with letters. The whiskers in **c**, **d** represent the range described by $[maximum(AUC - se, 0), minimum(AUC + se, 1)]$, where $se$ is the standard error calculated as $\frac{1}{\sqrt{n}}$, where $n$ is the number of samples used for this estimate ($n = 63, 52, 30$ and $22$ independent PBS for the disease detection disease classification, MDS genetic subtyping and anemia classification tasks, respectively).

morphotypes determined by Morphotype analysis were enriched in known cell types. Three hematologists labeled up to 1746 RBC and 1600 WBC. This demonstrates that morphotypes are enriched in known RBC and WBC types (Fig. 7a, b), but inter-expert concordance was limited for some rare cell types (particularly hypolobulated neutrophils and blasts; Supplementary Fig. S14). These results are also observed in single objective morphotype analysis, particularly for disease detection and classification models (Supplementary Fig. S15). Furthermore, the CMs enriched in artifacts were rarely enriched with known cell types.

To validate the accuracy and robustness of our models, we used a second cohort of 63 slides from the CUH2 cohort representing a similar spectrum of diagnoses to the MLL cohort but digitized using a

different slide scanner (Aperio AT2). In all cases, we evaluated the best performing fold from the previous cross-validations—we did this to get a clear measure of the real-world performance of such methods in a clinical context using the prediction of a single model with interpretable morphotypes, rather than a set of models which may not be available or yield slightly different results. Both our models—glmnet and multi-objective Morphotype analysis—displayed good generalization, with most external validation AUC intervals overlapping with cross-validated AUC estimates (Fig. 7c, d). We note that including morphotypes found to be statistically unstable in the original discovery step led to a deterioration of validation accuracy in the disease detection task. Finally, the single objective Morphotype analysis yielded worse generalization even when limiting to stable morphotypes

(Supplementary Fig. S16), indicating that simultaneously learning multiple tasks unravels more robust morphotypes.

## Discussion

We present an automated protocol for the detection and characterization of thousands of blood cells in PBS linked with machine-learning methods that can use these cellular descriptions to distinguish between clinical conditions and identify novel associations between cytomorphological phenotypes and clinical diagnoses. Importantly, we show that our approach generalizes to other centers and scanners.

Haemorasis, our open-source method to extract and characterize large numbers of WBC and RBC from digitized PBS, demonstrates how this can be automated with no recourse to proprietary software. We make it publicly available as a Docker container, enabling its straightforward application. Using Haemorasis, we detect and characterize over half a million WBC and millions of RBC. With morphometric moments (the mean and variance of morphometric features for each PBS) we show the diagnostic importance of cytomorphological heterogeneity for various conditions. This observation bears similarity to previous reports that associated RDW with increased AML transformation risk[42,45]. It is also worth considering that quantifying morphological variation, especially of subtle features, is likely to be challenging to achieve by visual assessment, as it requires the absolute quantification and evaluation of large numbers of cells.

To establish disease-associated cytomorphological changes of RBC and WBC, we developed morphotype analysis and applied it to over half a million WBC and millions of RBC. This showed that MDS cases with a larger prevalence of hypolobulated neutrophils and larger RBCs are more likely to harbor *SF3B1* mutations. The latter finding corroborates previous findings, where higher MCV was observed in *SF3B1*-mutant MDS when compared with other MDS subtypes[46,47], highlighting the role of *SF3B1* mutations in erythropoiesis[48] and the potential role of PBS RBC morphology in diagnosis. Additionally, neutrophil hyperlobulation was robustly detectable not only in MA but also in IDA[49,50], demonstrating that the ability to computationally analyze large numbers of cells can detect this feature even when it is subtle and would otherwise require enumeration of large numbers of neutrophils and their lobe count by experts[50]. We also observed larger neutrophils in MA, highlighting a common mechanism behind the enlargement of both RBCs and neutrophils in this condition. Reassuringly, morphotypes are enriched with known cell types and that our approach validates externally, generalizing to PBS from other centers digitized using different slide scanners. Nonetheless, further studies are required to validate the clinical relevance of the described morphotypes using more diverse training and validation data, to accommodate possible preparation- and scanner-specific artifacts[51,52]. Furthermore, while we determine new and previously known morphological trends, others pertaining to rarer cell types may require other approaches more sensitive to rarer cells. Here, we tried to maximize the explainability of our models by retrieving morphotypes and enhancing their generalization by using stable morphotypes, which eliminate unlikely solutions specific to subsets (folds) of the training data. However, the truth is that artificial intelligence models, more often than not, fail in clinical settings[53,54], such that the path forward should include the clinical application of these models as a point-of-care solution with larger sample sizes. Additionally, ongoing efforts to establish multicentre cohorts and tissue-bank for MDS, such as the National MDS Natural History Study[55], could be used as a better assessment of how well computational cytomorphology performs in routine MDS diagnosis.

It is also important to point out some additional limitations of our method. Firstly, the blood cell detection techniques, ranging from WBC and RBC detection to nuclear segmentation in WBC and morphometric characterization, could be improved. For example,

methods using transformers for hierarchical multi-resolution deep-learning architectures can replace the task of using predefined features, which can also bias results, as has been recently shown in histopathology[56]. Our protocol was designed to mimic the typical steps of hematological assessment—detecting relevant regions of the slide/image, identification of individual cells and analyzing their morphology. Other steps were pragmatic: owing to their great numbers, annotating RBCs in order to train a dedicated detection algorithm would be time consuming; for this reason, we opted for a fast and simple, but perhaps less accurate detection protocol. With appropriately annotated data, supervised RBC detection could be further improved with deep-learning models. Lastly, the characterization of blood cells and nuclei using self-supervised or unsupervised deep-learning methods might also help avoid some biases introduced by more predefined sets of features.

Notwithstanding the limitations discussed above, our work provides proof-of-principle that computational cytomorphology can augment the ability of automated blood cell analyses to identify abnormalities suggestive of hematological disease, with minimal additional cost. This can help identify patients needing further and usually more invasive and expensive testing, such as bone marrow aspirates or genomic sequencing. Recent applications of computational cytomorphology on bone marrow smears have demonstrated how it can identify leukocytes[31,57] and assist diagnostic predictions[28–30] in specialized haemato-oncology. By demonstrating that this can now be extended to blood smears/slides, our work reveals the potential for the large-scale incorporation of automated cytomorphology into routine diagnostic workflows.

## Methods

### Collection and digitalization of peripheral blood slides

Three retrospective sets of PBS with coverslips from two different centers were digitized at 40× magnification using two different slide scanners:

- Training/discovery
  - CUH1: used for training of cell detection. 54 PBS from randomly selected cases. PBS were automatically prepared at Cambridge University Hospitals (CUH) and scanned using a Hamamatsu NanoZoomer 2.0 (ndpi format).
  - MLL: used for training the disease prediction models and discovery of disease-associated morphologies (or computational morphotypes, as detailed below in the Methods). 362 PBS from individuals with MDS with mutations in either *SF3B1*, *SRSF2*, *U2AF1* or *RUNX1*, iron deficiency anemia (IDA), megaloblastic anemia (MA) and hematological controls. Manually prepared at Munich Leukemia Laboratory and scanned using a Hamamatsu NanoZoomer 2.0 (ndpi format).
- Validation – CUH2: used for validation of disease predictions. 68 PBS from individuals with MDS with mutations in either *SF3B1* or *SRSF2*, IDA, MA, and hematologically normal controls. The PBS were prepared manually (MDS) or automatically (controls, anemias) at CUH and digitized using an Aperio AT2 (svs format).

MLL data were collected and digitized with individual informed consent for research purposes and the study was reviewed and approved by the Munich Leukemia Laboratory's internal institutional review board and follow the European Union's General Data Protection Regulation (GDPR). Regarding Addenbrooke's data, the study was approved by the National Health Service Health Research Authority and the Health and Care Research Wales (Research Ethics Committee reference: 23/PR/0578).

We either digitized the entire PBS or selected the region of the PBS containing blood cells. Each slide was inspected and removed if lacking in quality (Supplementary Methods) and the final cohort composition

**Table 3 | Condition-specific composition of each cohort. Numbers in brackets represent the total after excluding poor quality digitized PBS**

| Condition | Cohorts (scanner) | | |
|---|---|---|---|
| | MLL (NZ2) | CUH1 (NZ2) | CUH2 (AT2) |
| Control | 58 | 54 | 11 |
| *SF3B1*-mutant MDS | 131 (130) | | 19 (16) |
| *SRSF2*-mutant MDS | 40 (38) | | 15 (14) |
| *RUNX1*-mutant MDS | 34 (33) | | |
| *U2AF1*-mutant MDS | 28 (26) | | |
| Megaloblastic anemia | 44 (40) | | 8 (7) |
| Iron deficiency | 27 | | 15 |
| Total | 362 (352) | 54 | 68 (63) |

Cohorts were retrieved from the Munich Leukaemia Laboratory (MLL) or from Cambridge University Hospitals (CUH1/2) using either a Hamamatsu Nanozoomer 2.0 (NZ2) or an Aperio AT2 (AT2).

is presented in Table 3. Details for MLL samples regarding age, sex, blood counts, and clinical diagnostic, and CUH2 regarding blood counts and clinical diagnostic are listed in Supplementary Table S1, Supplementary Table S2, respectively. To the best of our knowledge, no treatment had been administered at the time of sample collection and slide preparation.

## Haemorasis—computational detection and characterization of blood cells

Our blood cell detection and analysis pipeline—Haemorasis—assesses small, computationally tractable parts of the large PBS scans and consists of the following four steps (Supplementary Methods):

(1) **Quality control** to detect informative 512 × 512 pixel areas on each slide based on DenseNet121[58]. This removes tiles where the concentration of cells was too small (very few/no cells) or too large (high frequency of overlapping cells) or images were blurred (Supplementary Fig. S1). This ensures that the analyzed area corresponds approximately to the monolayer, the recommended area of analysis for hematologists[59].

(2) **Red blood cell detection** on tiles using a combination of Canny edge detection[60] and other simple computer vision operations and filtering of other objects (platelet clumps, groups of RBCs or individual WBCs) with an XGBoost model[36], using as features the morphometric characteristics described below in point **4.** and more extensively in the Supplementary Methods. In essence, we generated a set of candidate RBCs which were then categorized according to their morphometry as illustrated by other methods[61–66].

(3) **White blood cell detection** and segmentation from tiles was based on U-Net[35], a popular and robust algorithm for cell segmentation[67]. WBC nuclei were segmented by clustering WBC pixels using k-means clustering (k = 2)[68], assuming that the darker region of the WBC corresponds to the nucleus as shown by others[68].

(4) **Morphometric characterization of RBC and WBC** was performed using well-established morphometric features, available in popular bioimage analysis packages[37–39]. 53 features were calculated for each WBC (42 for cellular characterization and 11 for nuclear characterization) and 42 features for each RBC (Supplementary Methods).

## Cytomorphological prediction of clinical conditions

We assess the predictive performance of Haemorasis with four binary prediction tasks:

1. Disease detection—identifying the presence of either deficiency anemia or MDS vs. normal blood;

2. Disease classification—distinguishing between deficiency anemia and MDS;

3. MDS genetic subtyping: distinguishing between *SF3B1*-mutant and *SF3B1*-wildtype (*SF3B1*-wt) MDS;

4. Anemia classification—distinguishing between IDA and MA.

**Machine-learning using morphometric moments.** We assess the predictive performance of morphometric moments using elastic-net regression (glmnet)[40] with 5-fold cross-validation on MLL, calculating the cross-validated area under the receiver operating characteristic curve (AUC). Morphometric moments, the mean and variance of each feature for each cell type (RBC, WBC), were calculated for each PBS individually and used as a proxy for the distribution of each feature on each PBS. We test how blood counts (WBC counts (WBCC; cells/μL), hemoglobin concentration (Hb; g/dL) and platelet counts (Plt; platelets/μL)) affect classification performance, and preprocess features by standardizing them. Finally, we assess the contribution of features/groups of features on prediction (Supplementary Methods).

**Morphotype analysis.** The task of diagnosing hematological conditions from a PBS can be abstracted—given a set of objects (cells), each is classified into a given class (cell type) and the presence/relative prevalence of different cell types is indicative of specific hematological conditions. This can be viewed as a problem of multiple instance learning (MIL), a machine-learning field that focuses on classifying a set of objects based on its composition[69].

Considering this, we devised Morphotype analysis, an approach that (i) identifies relevant morphological classes of cells (computational morphotypes—CMs) without recourse to human-based cell annotation and (ii) distinguishes between conditions using CM proportions. Morphotype analysis can also incorporate other data types (i.e., blood counts; Supplementary Methods). We consider WBC and RBC separately, deriving separate WBC and RBC CMs. Being continuous, we optimized Morphotype analysis using gradient-descent (particularly Adam[70]). We test Morphotype analysis using a single set of CM for the four tasks specified above (multiple objectives—MO) and with a different model for each task (single objective—SO), and the impact of different assumptions regarding the number of CM (25 and 50 for MO and 10, 25, and 50 for SO) on prediction. For validation, we considered only stable CMs—less biased CMs which are consistently detected across different cross-validation folds (Supplementary Methods). Similarly to our models using morphometric moments, we tested the effect of blood counts on classification predictions.

## Validation

**Expert annotation of blood cells.** Three expert clinical hematologists annotated 1746 RBC and 1600 WBC automatically detected in MLL PBS to assess whether CMs were enriched in any expert-annotated blood cell type. RBC and WBC were annotated as belonging to a set of classes including normal and abnormal cell types and artefacts (Supplementary Methods). Enrichment for each CM was calculated as the ratio between the proportion of cells of a given expert-annotated cell type belonging to that CM (type/CM), divided by the proportion of cells of the given cell type in the entire set of expert-annotated cells (type/total).

**External validation.** To externally validate the performance of our predictive methodologies—glmnet with morphometric moments and Morphotype analysis—we tested the best performing models on CUH2, reporting their AUC estimates with standard errors (calculated as $\frac{1}{n}$, where $n$ is the size of the validation sample).

## Statistical analysis

All statistical analyses in this work were conducted using the R statistical software (v3.6.3)[71]. The MASS package[72] was used to calculate robust R and dunn.test[73] was used to calculate Dunn-Bonferroni tests.

Machine-learning models were implemented in either R (glmnet package[40]) or Python[74] for morphotype analyses (using PyTorch[75]).

## Reporting summary

Further information on research design is available in the Nature Portfolio Reporting Summary linked to this article.

## Data availability

The digitized PBS image data generated in this study and used for training (MLL) have been deposited in the BioImage Archive database under accession code S-BIAD440. The annotated datasets for tile quality classification, white blood cell segmentation and red blood cell filtering are available in https://doi.org/10.6084/m9.figshare.19153760. The machine-learning model parameters are available at https://doi.org/10.6084/m9.figshare.19164209. The necessary data to run Morphotype analysis is available at https://doi.org/10.6084/m9.figshare.19372292. The output of the Morphotype analysis, as well as the expert annotated cells, and the data necessary for downstream analysis are available at https://doi.org/10.6084/m9.figshare.19369391 and https://doi.org/10.6084/m9.figshare.19371008, respectively. An online platform for morphotype visualization is available in https://josegcpa.github.io/haemorasis-umap and the data supporting it is available in https://github.com/josegcpa/json-haemorasis.

## Code availability

We have made the Haemorasis pipeline available in https://github.com/josegcpa/haemorasis and as a Docker container in https://hub.docker.com/repository/docker/josegcpa/blood-cell-detection (Supplementary Methods). Morphotype analysis (mil-comori) and the statistical analysis and plot generation code (analysis-plotting) are available at[76]. The code for the quality control network is available at https://github.com/josegcpa/quality-net. The code for the U-Net is available at https://github.com/josegcpa/u-net-tf2.

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

## Acknowledgements

J.G.A. was supported by the NIHR Cambridge BRC and their opinions are not necessarily those of the NHS, the NIHR or the Department of Health and Social Care. G.S.V. is funded by a Cancer Research UK Senior Cancer Fellowship (C22324/A23015) and work in his lab is also funded by the European Research Council, Kay Kendall Leukaemia Fund, Blood

Cancer UK and the Wellcome Trust. We would like to acknowledge Drs. Martin Besser, James Russell, and Duncan Brian for their efforts in annotating blood cells.

## Author contributions

Contribution: M.G., G.S.V., and J.G.A. conceived the project and wrote the manuscript. J.G.A. developed and implemented the project. J.G.A., E.G., and J.C. scanned peripheral blood slides. T.H. provided peripheral blood slides and assisted with scanning and retrieval of blood count data for the Munich Leukemia Laboratory Set. W.G.D. retrieved blood count data. M.B. and E.G. annotated white blood cells.

## Funding

## Competing interests

G.S.V. is a consultant for Astrazeneca and STRM.BIO. The remaining authors declare no competing interests.

## Additional information

[1]European Molecular Biology Laboratory, European Bioinformatics Institute (EMBL-EBI), Hinxton, UK. [2]Department of Haematology, Cambridge Institute for Medical Research, University of Cambridge, Cambridge, UK. [3]Wellcome Sanger Institute, Wellcome Genome Campus, Cambridge, UK. [4]Munich Leukemia Laboratory GmbH, Munich, Germany. [5]Wellcome-MRC Cambridge Stem Cell Institute, University of Cambridge, Cambridge, UK. [6]Department of Haematology, University of Cambridge, Cambridge, UK. [7]Division of AI in Oncology, German Cancer Research Center (DKFZ), Heidelberg, Germany. [8]Present address: Champalimaud Foundation—Centre for the Unknown, Lisbon, Portugal. ✉e-mail: gsv20@cam.ac.uk; moritz.gerstung@dkfz.de

