## [Peer Review File · Nature Communications]

Reviewers' Comments:

Reviewer #1:

Remarks to the Author:

Thank you for the opportunity to review this manuscript by de Almeida et al.

The authors present results of Haemorasis, a machine learning protocol that allowed to capture morphological features of white cell and red cells and correlate that information with molecular features of a cohort of MDS samples.

I have to disclose that I am not expert on AI or other computational methods.

I can only judge relevance to clinical practice, potentially from a user perspective.

The paper basically describes the process of data acquisition, starting with scanning, and then the correlation analysis using this technology.

The authors find associations between morphological assessment of RBCs and WBCs with specific subsets of MDS. They do so by describing different morphotypes.

The conclusion is that this type of technology may guide in the evaluation and decision making of patients with peripheral blood alterations.

This is obviously a very interesting and detailed analysis of what is likely the future of morphological diagnosis.

That said, I have a couple of comments:

#1-the sample size is relatively small (I am not asking to increase the cohort) and these studies will obviously need to be expanded and confirmed.

#2-I find the paper extremely difficult to follow as it is written as a technical report. A lot of the data is in the supplemental information. This makes reading the paper very challenging.

#3-it should be easier to find detailed patient characteristics in the front of the paper. Indeed, the paper seems to miss detailed patient description including therapy and transfusion needs (if any). I would add a table with these data to the paper itself.

#4-Following on this:

a) were any of these patients red cell transfusion dependent?

b) were any of these patients treated?

c) If so, what was the effect of these "permutations" (treatment/transfusion) on data analysis and interpretation.

Thank you

Reviewer #2:

Remarks to the Author:

This manuscript presents a multi-step machine learning framework for blood cell detection and characterization in peripheral blood slide (PBS) images. The cellular descriptions can be used to predict diseases, e.g., myelodysplastic syndromes (MDS) or anemia, and identify novel associations between cytomorphological phenotypes and clinical conditions. The experiments show that SF3B1-mutant MDS can be separated from other MDS using cytomorphology and blood counts. The framework is also evaluated using image data acquired with a different image scanner to demonstrate its generalization ability.

Detailed comments:

1. The proposed framework consists of four major steps, i.e., quality control, red blood cell (RBC) detection, white blood cell (WBC) detection and morphometric characterization of RBCs and WBCs. For each step, an established algorithm is chosen for a specific task. However, the motivation of algorithm design/selection is not clearly explained, and the algorithm selection is not well verified. For instance, why a complicated model, DenseNet121, is selected for quality control, which seems to be less difficult than RBC detection/segmentation that is more complex but uses simple image processing techniques such as Canny edge detection? How to determine whether the selected algorithm is proper for specific tasks, given that many related algorithms can accomplish the same

task, e.g., quality control and cell detection/segmentation? It would be desirable to clearly explain and verify algorithm design in each step of the proposed framework.

2. Why are images with high frequency of touching or overlapping cells considered poor quality and needed to be excluded from subsequent analysis? Is it possible that the spatial distribution of those touching/overlapping cells can produce useful feature representations for disease diagnosis or prediction? It would be helpful to discuss this in the manuscript.

3. A Canny detection-based algorithm (i.e., Algorithm 1 in Supplementary Methods) is used for RBC detection/segmentation. However, there are many hyperparameters in the algorithm, such as area thresholds (line 221 in the supplementary material), contour thresholds (line 223) and ellipse thresholds (line 227). How to select optimal threshold values is not discussed, and how the threshold values affect the disease prediction is not clear. In addition, an XGBoost model is used to find true RBCs, but what features are used to represent RBC candidates is not clear.

4. In line 342 in the main text, the paper states "In all cases, we evaluated the best performing fold from the previous cross validations." Why only evaluate the best performing fold instead of reporting the average performance over the cross-validations?

5. What is the running time for the proposed framework to process a PBS image? In particular, the framework uses test-time augmentation for U-Net-based WBC detection/segmentation, which may be computationally expensive for high-dimensional PBS images but seems not to provide significantly better performance (but only slightly higher accuracy) than the counterpart without test-time augmentation, as shown in Figure 2b.

6. Given that the proposed framework consists of multiple non-trivial steps, some of which may need manual algorithm adaptation to different input images, the clinical value or application of the proposed framework needs further verification. It would be helpful to provide evidence that the framework could be deployed to routine diagnostic practice.

Reviewer #3:

Remarks to the Author:

Review of "Computational analysis of peripheral blood smears detects disease-associated cytomorphologies" by José Guilherme de Almeida et al.

General

The authors have studied the cytomorphology of peripheral blood smears from three different centers and digitized two different scanners. They employed both conventional and deep learning-based methods. As a remark, the method section is a great example on how to describe both clearly and in detail the used techniques – this applies also for the Supplementary section. Computer vision has not been used as much in hemato-oncology as in H&E-stained samples of solid tumors implying that this work will act as a reference for future publication in this field.

In summary, the work is technically sound, the dataset sufficient and the results fascinating. As I found the work interesting, I have many, yet only minor revisions to ask to improve the presentation of the work.

Abstract

L19 "via a range of cytopenias and dysplastic changes of blood and bone marrow cells"
- Suggestion. "via cytopenias, blast expansion, dysplasia of hematopoietic cells"

L22 "molecular testing" is a bit abstract. Do you mean "molecular genetics testing"?

L28 "diagnosis and prognosis"

- The work did not directly study prognosis. Please reformulate.

L29-31 "We find that hypolobulated neutrophils and large RBC are characteristic of SF3B1-mutant MDS, and, while prevalent in both iron deficiency and megaloblastic anemia, hyperlobulated neutrophils are larger in the latter."

- Consider splitting the sentence into two.

Introduction

L36-37 "For example, anemias, characterized by reduced hemoglobin concentration (Hb) and red blood cell (RBC) numbers"

- Not entirely true. RBC count can be high in thalassemias.

L39-42 "For this reason, the diagnosis of MDS requires the detection of cytopenias, of changes to white blood cell (WBC) and RBC maturation blood cell through the analysis of cytomorphology in bone marrow (BM) and peripheral blood slides (PBS), cyto- and histochemistry, karyotyping and immunophenotyping⁴⁻⁸."

- Not correct. The diagnosis is based on morphological evidence of dysplasia. Other findings presented here are complementary or used in prognostics.

L45-46 "the treatment of MDS generally involves chemotherapeutic agents blood/platelet transfusions¹³."

- Correct, but please also mention HMAs.

L47-48 "molecular characterization"

- Replace with "molecular genetics" or something more appropriate

Materials and methods

L74-85

- Please indicate the scanner magnification, resolution use of dry or oil immersion and the exported image formats
- Were the whole slides digitized with all scanners or just representative areas?

L74-85 "CUH1: used for training of cell detection" ... "MLL: used for training and discovery of disease-associated morphologies." ... "Validation – CUH2: used for validation of disease predictions."

- Please clarify. Are disease-associated morphologies and disease prediction the same? If yes, use just either term. Was cell detection validated in the other cohorts?

L79 "SF3B1, SRSF2, U2AF1 or RUNX1, IDA"

- What is IDA?

L83-85 "The PBS were prepared manually (MDS) or automatically (controls, anemias) at CUH and digitized using an Aperio AT2."

- Is it possible that differences in slide preparation affect the results?

L87-89 "Details on MLL regarding age, sex, blood counts and clinical diagnostic, and CUH2 regarding blood counts and clinical diagnostic are available in Supplementary Table S1 and Supplementary Table S2, respectively."

- Please indicate any differences in these parameters between the cohorts.

L101-103 Why did you not use DL-based methods for RBC detection as you did for quality control and WBC detection? I assume that the Canny edge detection was perhaps faster to generate but not necessarily faster or better to run on the dataset. In addition, I believe that non-parametric methods would be better in terms of generalizing over other datasets and identifying RBC variants. I am afraid that this method is the weak point of your pipeline as it could exclude abnormal RBC variants (for instance teardrop RBCs, rouleaux). NB! I do not ask to repeat the analysis with a DL-based method but to further explain your decisions.

L104-105 Why did the authors use U-Net and not faster R-CNN for WBC segmentation? Faster R-

CNN takes cell morphology into account and should lead to better segmentation results.

L104-105 Identification of WBCs using k-means clustering seems very risky as cells might have dark membrane folds, artefacts, high granularity etc. Did you experience any challenges? Could you add examples of successful and unsuccessful nuclei segmentation? Especially the results of eosinophils and basophils (high granularity) and darker cells would be interested to see. Perhaps an instance segmentation algorithm such as faster R-CNN could be more robust.

L133-134 "Being fully differentiable, we optimize Morphotype analysis using gradient-descent (particularly Adam41)."

- What does "fully differentiable" mean?

L138 "across different folds"

- You mean the folds of crossvalidation or something else?

Results

General comment. Add p values to all boxplots and scatter plots to help understand which differences are significant.

Fig 1. Add p values between cohorts within the plot or to a separate table

Fig 1b

- Replace the y-axis with the proportion instead of the number. So the proportion of female normal + male normal = 100%; female MDS + male MDS = 100%; female normal + male anaemia = 100%

L192 "This (i) reduces the detecting non-cellular"

- This (i) reduces the detection of non-cellular

L193 "the interpretable part"

- Reformulate for example "the clinically interpretable section"

L195-199 "To detect WBC, we trained a U-Net-based³⁴ DL model with extensive data augmentation (random image alterations which make models more robust) during training on a dataset with >2,800 manually annotated WBC in PBS from CUH1 and validated this on testing sets from CUH1, CUH2 and the MLL cohort, with prediction post-processing and test time augmentation (TTA) improving predictions (Figure 2b, Supplementary Figure S2a-b)."

- Split the sentence to improve readability.

L199-200 "We confirmed the good performance of the model through visual inspection"

- I agree but isn't the Supplementary Figure S2a-b also a quantitative metric of the model? Please add also the mean detection accuracy and recall or AUC values in the test set.

L204-205 Please explain where the "1 in 7 to 1 in 300" proportions originate.

Fig. 2e. Add statistics (p values and mean or median values)

Fig. 2f. Add statistics (p and correlation value)

L249 "the distribution of cytomorphological"

- Consider replacing with "the two-dimensional UMAP distribution of cytomorphological" or similar

L260 "producing human-interpretable, disease-associated cytomorphologies"

- Can you show that these are human-interpretable? The authors should prepare static or interactive UMAPs clustered by CMs to visualize example images of individual CMs with different parameters (10, 25, 50). Hence, the reader could interpret any associations with cell types.

Fig. 4

- This is probably the most important results of the manuscript. What I find not surprising but a bit disappointing is that most cells are related to neutrophils and lymphocytes = the two most common WBCs in the PBS. Furthermore, you even find two CMs that are very close to each other (WCM-1 and WCM-3 are quite similar as well as WCM-2 and WCM-4). I suspect that the approach cannot find robust morphotypes of rarer cells such as other granulocytes, which are known to harbor dysplastic patterns in MDS. Could you please explain why the non-stable MCs are not integrated in the analysis? I would not expect that there should all morphotypes between MDS, anemia and normal should be stable.

Even though I appreciate the novelty of the approach, I do not agree that this will solve well the cytomorphological composition of the distinct diseases in question. Therefore, the limitations of this approach should be discussed.

Fig. 5

- This is really fascinating. No critical comments to share.

L325 "by expert hematologists"

- Small detail, but were these really clinical hematologists or perhaps rather hematopathologists or laboratory physicians?

Fig. 6

- The legend for figures c and d requires further clarification that the bars are from the training data and whiskers from the validation data

L345-346 "We note that including morphotypes found to be statistically unstable in the original discovery step led to a deterioration of validation accuracy in the disease detection task"

- Why do you think this happens? Provide some explanation in the Discussion part

Discussion

I find it a bit surprising that the authors did not use the modern approach where visual features would be extracted from the image tiles or segmented cells to predict the phenotype of interest (in your case 4 binary classifications). This would have allowed you to explore the morphologies with a less supervised approach and visualize these with GradCAM etc. Rather, the authors crafted explainable features (partly with using DL methods) and associated these with the disease phenotypes. I would suggest the authors to discuss these two technical approaches and argument why did you chose the other approach.

L369-372 "Additionally, neutrophil hyperlobulation was robustly detectable not only in MA but also in IDA55,56 demonstrating that the ability to computationally analyze large numbers of cells can detect this feature even when it is subtle and would otherwise require enumeration of large numbers of neutrophils and their lobe count by experts56."

- I wonder whether the IDA and MA patients developed any hematological malignancy in the next 5 years? PBS are not routine samples for IDA and MA. Therefore, please confirm that these patients were not associated with hematological malignancy.

Supplementary methods

L134-136 Considering that this may lead to detecting the same cell twice as the same object, we exclude a prediction if a cell has already been predicted within 8 pixels.

- Why 8 pixels?

L175 "from the MLLS"

- MLL?

L182 Why 25 epochs? Why not training with callbacks?

L183 "every time"

- You mean after every epoch?

L181-182 "Tiles were artificially augmented (computationally altered) during training through random flips/rotations/JPEG compression."

L184-186 "During training, each image had a 60% probability of having its brightness, saturation, hue and contrast randomly altered (by 15%, 10%, 10% and 10%, respectively) or of having random JPEG compression artefacts introduced."

- Am I missing something or are the lines 181-182 and 184-186 contradictory to each other?

L167 Quality control network

- I would be interested to see if there are any differences in the amount of good vs. poor quality tiles a. per cohort and b. per manually vs. automatically prepared slides.

L198 You excluded groups of RBCs. Do you think it is possible that these could share biological information that individual RBCs would not?

L257-259 Data augmentation and Supplementary Table S3

- did you manually validate that the augmented data were realistic representations of the true data? This should be confirmed to the supplementary material.

L266-268 "Prediction post-processing is done by removing objects detected as WBC whose size lies outside the expected distribution for WBC sizes, and filling small convex hull defects."

- How did you define these reference values?

Supplementary Figure S2

- Why is the IoU substantially lower in CUH2 with depth mult. 0.25 than in CUH1 and MLL?

L276-278 To account for different resolutions (0.2268 micrometers/pixel for Hamamatsu NanoZoomer 2.0; 0.2517 micrometers/pixel for Aperio AT2) we rescale the images in CUH2 by a factor of 1.1098 prior to cellular characterization.

- I have difficulties to understand this. You might be right in doing this but please explain more why this is needed. Won't this inflict a bias on the cell perimeter?

Supplementary Figure S5

- I find the figure, especially A and C so important that these should be in the main figures.

- Did you normalize the features for glmnet? If not then, the coefficient (NB! Correct the x-axis label) does not reflect feature importance!

- Age, gender and cytopenias are known to associate with bone marrow morphology (Brück et al Blood Cancer Discovery 2021). Based on Fig 1, the cohorts differed by these factors. Please show, if you can predict age, cytopenias and gender with the features (if feasible, separately in normal, MDS and anaemia patients). If yes, then the authors should exclude these features from the other models to find true age- and gender-independent features.

Rebuttals to reviewer comments

We would like to start by expressing our appreciation of the work done by reviewers for this work and by thanking them for having taken the time to review our manuscript and for providing helpful and relevant suggestions. Below we provide, by the order determined by the editor and reviewers, the reviewer comments and our responses to them in blue. In the main text and supplementary information we have provided the noted corrections and have signaled them with gray highlights. The only changes made to the reviewer's comments are relative to the relevant separation of comments into relevant sections, having left everything else unchanged and in black.

In short:

- For the remainder of this document: in black text are the reviewer's comments and in blue text are the rebuttals from the authors.
- For the manuscript and supplementary information: changes are noted in gray highlights.

We also mention the lines at which changes were made. Unless otherwise noted, these lines refer to the main text; whenever lines in the SI are relevant, they are explicitly mentioned (i.e. "lines xx-yy in the SI").

Reviewer #1

Thank you for the opportunity to review this manuscript by de Almeida et al.

The authors present results of Haemorasis, a machine learning protocol that allowed to capture morphological features of white cell and red cells and correlate that information with molecular features of a cohort of MDS samples.

I have to disclose that I am not expert on AI or other computational methods.

I can only judge relevance to clinical practice, potentially from a user perspective.

The paper basically describes the process of data acquisition, starting with scanning, and then the correlation analysis using this technology.

The authors find associations between morphological assessment of RBCs and WBCs with specific subsets of MDS. They do so by describing different morphotypes.

The conclusion is that this type of technology may guide in the evaluation and decision making of patients with peripheral blood alterations.

This is obviously a very interesting and detailed analysis of what is likely the future of morphological diagnosis.

We thank the reviewer for their positive assessment of our work.

That said, I have a couple of comments:

1. The sample size is relatively small (I am not asking to increase the cohort) and these studies will obviously need to be expanded and confirmed.

We do recognize that the sample size is relatively small when compared with other histopathology works (particularly those stemming from large consortia such as PCAWG¹), and for this reason we avoid drawing any grandiose claims about the applicability that can stem from this work. In particular, highlighting instead these results as a proof-of-concept for the utility of computational methods in the scientific and clinical analysis of computational methods. We now make a more explicit mention of this in the Discussion (line 365).

2. I find the paper extremely difficult to follow as it is written as a technical report. A lot of the data is in the supplemental information. This makes reading the paper very challenging.

We recognise some parts may be too brief and too quick to redirect the reader to the Supplementary Information, making the reading of this manuscript more complicated. We have added some information to different sections in order to avoid having to constantly refer to the Supplementary Information.

3. It should be easier to find detailed patient characteristics in the front of the paper. Indeed, the paper seems to miss detailed patient description including therapy and transfusion needs (if any). I would add a table with these data to the paper itself.

See response to point 4.

4. Following on this:
 - a. Were any of these patients red cell transfusion dependent?
 - b. Were any of these patients treated?

- c. If so, what was the effect of these “permutations” (treatment/transfusion) on data analysis and interpretation.

We thank our reviewer for points 3 & 4, which are entirely valid. We now clarify that samples were obtained at the time of initial diagnosis, such that patients had not started on any drug treatment as yet. We cannot rule out the possibility that occasional patients may have had a blood or platelet transfusion that was not documented, but believe that this was very uncommon. We have now clarified this in Methods (lines 91-92).

Reviewer #2

This manuscript presents a multi-step machine learning framework for blood cell detection and characterization in peripheral blood slide (PBS) images. The cellular descriptions can be used to predict diseases, e.g., myelodysplastic syndromes (MDS) or anemia, and identify novel associations between cytomorphological phenotypes and clinical conditions. The experiments show that SF3B1-mutant MDS can be separated from other MDS using cytomorphology and blood counts. The framework is also evaluated using image data acquired with a different image scanner to demonstrate its generalization ability.

We thank the reviewer for their time and for their succinct summary of our work.

Detailed comments:

1. The proposed framework consists of four major steps, i.e., quality control, red blood cell (RBC) detection, white blood cell (WBC) detection and morphometric characterization of RBCs and WBCs. For each step, an established algorithm is chosen for a specific task. However, the motivation of algorithm design/selection is not clearly explained, and the algorithm selection is not well verified. For instance, why a complicated model, DenseNet121, is selected for quality control, which seems to be less difficult than RBC detection/segmentation that is more complex but uses simple image processing techniques such as Canny edge detection? How to determine whether the selected algorithm is proper for specific tasks, given that many related algorithms can accomplish the same task, e.g., quality control and cell detection/segmentation? It would be desirable to clearly explain and verify algorithm design in each step of the proposed framework.

It is true that several algorithm choices were taken throughout this work. Particularly, DenseNet121 for quality control was chosen due to its speed and relatively good performance, comparable with other state-of-the-art methods. As for the RBC segmentation protocol, we focused on obtaining a heuristic method that would allow us to detect a large number of non-overlapping RBCs. We note that RBCs are relatively simple to detect (especially after using our quality control step to select appropriate regions of the image) and other publications make use of relatively simple methods to generate a set of candidate RBC, which are then refined with recourse to other methods (for example please see ²⁻⁷). We have made these choices clearer in the Supplementary Methods (lines 200-202 of the SI) and in the Methods (lines 105-107, 108-111). We also note that Supplementary Methods lines 169-174 of the SI provide an explanation for the choice of deep learning models for quality control.

2. Why are images with high frequency of touching or overlapping cells considered poor quality and needed to be excluded from subsequent analysis? Is it possible that the spatial distribution of those touching/overlapping cells can produce useful feature representations for disease diagnosis or prediction? It would be helpful to discuss this in the manuscript.

While it is indeed true that clusters of RBC can be features of diseases (e.g “rouleaux” formations or “coinstacks” and red cell clumping seen in cold-agglutinin disease ^{8,9}), when these are seen outside the monolayer region, they are frequently an artifact of slide preparation. For example, increased cell overlap can arise from spreading the same volume of blood over a shorter distance on the slide. Some of these problems can be overcome through the use of automated (machine) spreaders, but these are not used by many labs. Also, we wanted our findings to be relatable to practicing hematologists/diagnosticians, who typically analyse the thin layer of non-overlapping blood cells to make diagnoses. We have added a comment to the Supplementary Methods to further clarify this (lines 212-218 of the SI).

3. A Canny detection-based algorithm (i.e., Algorithm 1 in Supplementary Methods) is used for RBC detection/segmentation. However, there are many hyperparameters in the algorithm, such as area thresholds (line 221 in the supplementary material), contour thresholds (line 223) and ellipse

thresholds (line 227). How to select optimal threshold values is not discussed, and how the threshold values affect the disease prediction is not clear. In addition, an XGBoost model is used to find true RBCs, but what features are used to represent RBC candidates is not clear.

Our reviewer is correct in noting the number of hyperparameters in the RBC detection protocol. These were determined heuristically by inspecting the distribution of each contour in the dataset used for training the XGBoost model. As regards features used to find RBCs, we have characterized the RBC using the same features used later to classify disease. We have now made both points (the selection of thresholds + features used to train XGBoost) more explicit in the text (lines 200-202 in the SI and lines 104-107 in the main text).

4. In line 342 in the main text, the paper states “In all cases, we evaluated the best performing fold from the previous cross validations.” Why only evaluate the best performing fold instead of reporting the average performance over the cross-validations?

In this case, we chose to consider a less flexible evaluation of our models – how exactly would they perform in a clinical application context, where a single model would be used for prediction without recourse to a larger cohort of models. We have clarified this in the text (lines 323-325 and 361-363).

5. What is the running time for the proposed framework to process a PBS image? In particular, the framework uses test-time augmentation for U-Net-based WBC detection/segmentation, which may be computationally expensive for high-dimensional PBS images but seems not to provide significantly better performance (but only slightly higher accuracy) than the counterpart without test-time augmentation, as shown in Figure 2b.

After some tests, we came to the conclusion that the framework takes between 30 minutes to a few hours for each PBS (details added to Supplementary Results). This is in part, and as the reviewer correctly notes, due to TTA; but this is largely because we detect/analyze large numbers of cells in order to maximize performance of our approach. In fact, in some cases the number of detected cells surpasses 200 (the number recommended for blood smear analyses from the World Health Organization for MDS diagnosis ¹⁰) by 3 or 4 orders of magnitude. While improvements on the throughput of this approach would certainly be advantageous, we note that for our application and as a proof of concept, we chose to use the method that could best segment cells to avoid misdetections or introduction of other forms of artifacts. For this reason, while TTA leads to relatively small improvements, we chose it in preference. We have now made this clearer in the text, offering some throughput metrics as well and analysis of this in the Supplementary Results (lines 506-530 of the SI).

6. Given that the proposed framework consists of multiple non-trivial steps, some of which may need manual algorithm adaptation to different input images, the clinical value or application of the proposed framework needs further verification. It would be helpful to provide evidence that the framework could be deployed to routine diagnostic practice.

We agree that the practical clinical assessment of our framework in the clinic would be the ideal case scenario. However, such an endeavor would require the collaboration of a sufficient number of clinicians/diagnostic laboratories, the installation of new equipment (scanning microscopes), the coordination of a server service where these GPU-dependent computations could be performed and in many cases the approval of the process by research and development offices and ethics committees. This is not a trivial undertaking and goes beyond the aims of our manuscript, which is to show the potential of computational cytomorphology in the diagnosis of blood cancers. Nevertheless, we acknowledge the relevance of our reviewer’s comment to legitimate consideration that readers may have and have now added a suggestion of how future works can use similar frameworks to construct clinically useful tools to the Discussion, highlighting the caveats in our own approach (lines 357-359, lines 363-365, 369-380).

Reviewer #3

Review of “Computational analysis of peripheral blood smears detects disease-associated cytomorphologies” by José Guilherme de Almeida et al.

General

The authors have studied the cytomorphology of peripheral blood smears from three different centers and digitized two different scanners. They employed both conventional and deep learning-based methods. As a remark, the method section is a great example on how to describe both clearly and in detail the used techniques – this applies also for the Supplementary section. Computer vision has not been used as much in hemato-oncology as in H&E-stained samples of solid tumors implying that this work will act as a reference for future publication in this field.

In summary, the work is technically sound, the dataset sufficient and the results fascinating. As I found the work interesting, I have many, yet only minor revisions to ask to improve the presentation of the work.

We thank the reviewer for the very positive assessment of our work and for highlighting the effort we put into the Methods/Supplementary information. We are also grateful for their extremely helpful comments, which must have taken a lot of their time.

Abstract

L19 “via a range of cytopenias and dysplastic changes of blood and bone marrow cells”

- Suggestion. “via cytopenias, blast expansion, dysplasia of hematopoietic cells”

Thank you - we have included this suggestion (lines 18-19).

L22 “molecular testing” is a bit abstract. Do you mean “molecular genetics testing”?

Yes, we did mean molecular genetics testing and have further clarified this to avoid confusion (lines 21-22).

L28 “diagnosis and prognosis”

- The work did not directly study prognosis. Please reformulate.

We have replaced “prognosis” with “disease subtyping” to better reflect the presented results (line 26).

L29-31 “We find that hypolobulated neutrophils and large RBC are characteristic of SF3B1-mutant MDS, and, while prevalent in both iron deficiency and megaloblastic anemia, hyperlobulated neutrophils are larger in the latter.”

- Consider splitting the sentence into two.

We have rephrased the sentence to improve readability (lines 27-29).

Introduction

L36-37 “For example, anemias, characterized by reduced hemoglobin concentration (Hb) and red blood cell (RBC) numbers”

- Not entirely true. RBC count can be high in thalassemias.

We have clarified this (line 34).

L39-42 "For this reason, the diagnosis of MDS requires the detection of cytopenias, of changes to white blood cell (WBC) and RBC maturation blood cell through the analysis of cytomorphology in bone marrow (BM) and peripheral blood slides (PBS), cyto- and histochemistry, karyotyping and immunophenotyping⁴⁻⁸."

- Not correct. The diagnosis is based on morphological evidence of dysplasia. Other findings presented here are complementary or used in prognostics.

We have clarified this statement (line 37).

L45-46 "the treatment of MDS generally involves chemotherapeutic agents blood/platelet transfusions¹³."

- Correct, but please also mention HMAs.

We have added HMAs (line 44).

L47-48 "molecular characterization"

- Replace with "molecular genetics" or something more appropriate

We have made the suggested change (line 46).

Materials and methods

L74-85

- Please indicate the scanner magnification, resolution use of dry or oil immersion and the exported image formats

We have now added this information to the manuscript in the Methods section (line 71).

- Were the whole slides digitized with all scanners or just representative areas?

We have scanned the whole slide as we wanted the computational algorithms to do most of the heavy lifting and reduce the burden on collaborators and hematologists. We have clarified this in the Methods section (87).

L74-85 "CUH1: used for training of cell detection" ... "MLL: used for training and discovery of disease-associated morphologies." ... "Validation – CUH2: used for validation of disease predictions."

- Please clarify. Are disease-associated morphologies and disease prediction the same? If yes, use just either term. Was cell detection validated in the other cohorts?

We did validate cell detection across cohorts. By "disease predictions" we refer to the binary classification output (i.e. disease vs. control), whereas by "discovery of disease-associated morphologies" we refer to the visual and computationally-derived examples of disease-relevant cellular morphologies. We have now clarified this in Methods (in particular, we have added a reference to a later part of the methods to avoid excessive repetition; lines 77-78).

L79 "SF3B1, SRSF2, U2AF1 or RUNX1, IDA"

- What is IDA?

We use IDA as an acronym for iron deficiency anemia. We have now included the missing reference to this (lines 79-80).

L83-85 "The PBS were prepared manually (MDS) or automatically (controls, anemias) at CUH and digitized using an Aperio AT2."

- Is it possible that differences in slide preparation affect the results?

This is an excellent point that we also considered and were relatively concerned about. We believe that our results in the external validation dataset, together with the performance of the segmentation algorithm across cohorts prove that our method is, at least to a significant extent, unaffected by this. However, a more significant assessment was necessary - we now note in the Results that the number of cells and the area of the region that is of good quality depends on the cohort of origin and, in some cases, to the type of slide preparation (manual vs. automated; lines 216-224).

L87-89 "Details on MLL regarding age, sex, blood counts and clinical diagnostic, and CUH2 regarding blood counts and clinical diagnostic are available in Supplementary Table S1 and Supplementary Table S2, respectively."

- Please indicate any differences in these parameters between the cohorts.

We have now included an additional table (Table 2) detailing the differences between both cohorts and discuss these differences accordingly in the Results (Table 2).

L101-103 Why did you not use DL-based methods for RBC detection as you did for quality control and WBC detection? I assume that the Canny edge detection was perhaps faster to generate but not necessarily faster or better to run on the dataset. In addition, I believe that non-parametric methods would be better in terms of generalizing over other datasets and identifying RBC variants. I am afraid that this method is the weak point of your pipeline as it could exclude abnormal RBC variants (for instance teardrop RBCs, rouleaux). NB! I do not ask to repeat the analysis with a DL-based method but to further explain your decisions.

We do appreciate that there was room for a deep-learning based solution for RBC detection, and that there are precedents for this in the literature. However, we were, at the time, hindered by a lack of annotated data; this would have been significantly more laborious to label than WBC as RBC density is considerably higher. For this reason, we chose to place a heavier emphasis on the quality control and RBC filtering steps – the segmentation, which can be done with relatively simple methods as noted in other works²⁻⁷, is then a heuristic method to generate a set of candidate RBC which are then filtered according to their likelihood. To reply to this point, as well as to the first point of reviewer #2, we have now made these choices and caveats much clearer in the manuscript, including in Methods (lines 105-107), Supplementary Methods (lines 200-202), and Discussion (lines 375-378).

L104-105 Why did the authors use U-Net and not faster R-CNN for WBC segmentation? Faster R-CNN takes cell morphology into account and should lead to better segmentation results.

Faster RCNN is only capable of detecting, rather than segmenting cells. Other options which combine detection and segmentation – particularly Mask RCNN – were certainly a relevant option, but given the popularity of U-Net models for bioimaging problems we decided to use this model. Additionally, there is evidence supporting the choice of U-Net-based solutions for cell segmentation – the winning solution for the 2018 Data Science Bowl, a competition to deliver the best algorithm at segmenting cell nuclei, is based on U-Net models¹¹. We have added a comment regarding this to the Methods section (lines 108-109).

L104-105 Identification of WBCs using k-means clustering seems very risky as cells might have dark membrane folds, artefacts, high granularity etc. Did you experience any challenges? Could you add examples of successful and unsuccessful nuclei segmentation? Especially the results of eosinophils and basophils (high granularity) and darker cells would be interested to see. Perhaps an instance segmentation algorithm such as faster R-CNN could be more robust.

We acknowledge that this segmentation is harder for the instances mentioned by our reviewer (eosinophils, basophils and cases of cells with low contrast due to high similarity between cytoplasm and nucleus). We have now included a Supplementary Figure (Supplementary Figure S4) highlighting some failure cases and mention these shortcomings in Results (lines 233-236), as well as discussing them in Discussion (lines 369-371). We particularly highlight how much more robust a deep-learning segmentation model could be for these cases.

L133-134 “Being fully differentiable, we optimize Morphotype analysis using gradient-descent (particularly Adam41).”

- What does “fully differentiable” mean?

Here we mean to say that the model as defined by us is *continuously* differentiable and, for this reason, it is possible to calculate a gradient for every single real-value point. We have replaced “fully” with “continuous” to make this more evident (line 140).

L138 “across different folds”

- You mean the folds of crossvalidation or something else?

We meant across folds of cross-validation. We have changed the text to “cross-validation folds” to make it clearer (line 145).

Results

General comment. Add p values to all boxplots and scatter plots to help understand which differences are significant.

Fig 1. Add p values between cohorts within the plot or to a separate table

Thank you, to address this a table (Table 2) has been added detailing all statistical tests, coefficients and p-values for this subsection of the Results.

Fig 1b

- Replace the y-axis with the proportion instead of the number. So the proportion of female normal + male normal = 100%; female MDS + male MDS = 100%; female normal + male anaemia = 100%

We agree that this works better and have altered the figure to reflect these changes (Fig. 1).

L192 “This (i) reduces the detecting non-cellular”

- This (i) reduces the detection of non-cellular

We have now corrected this (line 200).

L193 “the interpretable part”

- Reformulate for example “the clinically interpretable section”

We have now corrected this (lines 201-202).

L195-199 “To detect WBC, we trained a U-Net-based³⁴ DL model with extensive data augmentation (random image alterations which make models more robust) during training on a dataset with >2,800 manually annotated WBC in PBS from CUH1 and validated this on testing sets from CUH1, CUH2 and the

MLL cohort, with prediction post-processing and test time augmentation (TTA) improving predictions (Figure 2b, Supplementary Figure S2a-b).”

- Split the sentence to improve readability.

We have now split this sentence into three separate sentences for clarity (lines 203-207).

L199-200 “We confirmed the good performance of the model through visual inspection”

- I agree but isn’t the Supplementary Figure S2a-b also a quantitative metric of the model? Please add also the mean detection accuracy and recall or AUC values in the test set.

While 2a-b are quantitative analysis of the results, the purpose of 2c-d is for visual, qualitative inspection. We have added mean detection precision and recall. We did not use mean detection accuracy as the absence of a “true negative” makes the accuracy an unsuitable metric for this (Supplementary Figure 2, particularly panel c; lines 205-206 of the main text).

L204-205 Please explain where the “1 in 7 to 1 in 300” proportions originate.

We have made this clearer by explaining the calculation and opting for percentages rather than proportions. We also detected an imprecision in our calculations (instead of using the FPR we were using the FNR) which reduces the denominator (rather than 1 in 300 it should be closer to 1 in 50); we apologize for this and note that it still is a significant reduction in the number of false positives (lines 213-215).

Fig. 2e. Add statistics (p values and mean or median values)

We have now included a much more complete analysis of these numbers. However, we avoid directly comparing the total number of detected cells as this is dependent on the tiles predicted as being of good quality; for this reason, we have added to Results a section which focuses on this analysis (lines 216-222), together with a table showing the number of detected cells and cellular density across cohorts and conditions (Table 4). To avoid creating confusion between the image (absolute cell counts) and the analysis (cellular density) we avoid altering this table and include a new table instead (as mentioned).

Fig. 2f. Add statistics (p and correlation value)

We have now added this to the figure.

L249 “the distribution of cytomorphological”

- Consider replacing with “the two-dimensional UMAP distribution of cytomorphological” or similar

We have altered this to make it clearer that one is observing a two-dimensional representation of the data (lines 258-259).

L260 “producing human-interpretable, disease-associated cytomorphologies”

- Can you show that these are human-interpretable? The authors should prepare static or interactive UMAPs clustered by CMs to visualize example images of individual CMs with different parameters (10, 25, 50). Hence, the reader could interpret any associations with cell types.

To address this, we have prepared an online platform where the reader can visualize the examples of all CMs for the relevant model (25 CM + blood counts), available in <https://.github.io/haemorasis-umap> (lines 269-271; implementation details in lines 304-310 of the SI).

Fig. 4

- This is probably the most important results of the manuscript. What I find not surprising but a bit disappointing is that most cells are related to neutrophils and lymphocytes = the two most common WBCs in the PBS. Furthermore, you even find two CMs that are very close to each other (WCM-1 and WCM-3 are quite similar as well as WCM-2 and WCM-4). I suspect that the approach cannot find robust morphotypes of rarer cells such as other granulocytes, which are known to harbor dysplastic patterns in MDS. Could you please explain why the non-stable MCs are not integrated in the analysis? I would not expect that there should all morphotypes between MDS, anemia and normal should be stable.

Even though I appreciate the novelty of the approach, I do not agree that this will solve well the cytomorphological composition of the distinct diseases in question. Therefore, the limitations of this approach should be discussed.

The calculation of stability is a necessary step of this analysis as it allows us to determine which cell types are consistent across folds (as explained in the Methods and Supp. Methods). Without this, the computational morphotypes would be dependent on the subset of the data used on a given fold, making the morphotypes dependent on the best performing fold. Given that folds perform similarly, it would be unwise to proceed with the selection of all morphotypes present in a specific fold as some are simply not captured by most of the remaining folds. Nonetheless, we now discuss the limitations of this approach in Discussion (lines 359-363).

Fig. 5

- This is really fascinating. No critical comments to share.

Thank you!

L325 "by expert hematologists"

- Small detail, but were these really clinical hematologists or perhaps rather hematopathologists or laboratory physicians?

Yes, these were clinical hematologists (i.e. trained as diagnostic and clinical hematologists, rather than hematopathologists). We have now changed this to "clinical hematologists" (line 149 and 312).

Fig. 6

- The legend for figures c and d requires further clarification that the bars are from the training data and whiskers from the validation data

Thank you. We agree and have now clarified this.

L345-346 "We note that including morphotypes found to be statistically unstable in the original discovery step led to a deterioration of validation accuracy in the disease detection task"

- Why do you think this happens? Provide some explanation in the Discussion part

In essence, including non-stable morphotypes would lead to the inclusion of morphotypes which were detected in a single fold. While for each fold, we detect morphotypes that are likely to be relevant to the classification, they may be spurious, i.e. their detection may be an artifact of the subset of data present during training. In other words, some morphotypes may be inferred purely because they are more prevalent in a subset in a way that is not representative of the broader population of cells. We have included an explanation of this in Discussion (lines 361-363). We also note that the explanation provided in lines 371-375 of the SI also sought to clarify this.

Discussion

I find it a bit surprising that the authors did not use the modern approach where visual features would be extracted from the image tiles or segmented cells to predict the phenotype of interest (in your case 4 binary classifications). This would have allowed you to explore the morphologies with a less supervised approach and visualize these with GradCAM etc. Rather, the authors crafted explainable features (partly with using DL methods) and associated these with the disease phenotypes. I would suggest the authors to discuss these two technical approaches and argument why did you chose the other approach.

We have reformulated parts of the discussion to include this critical aspect of our analysis. We are aware that the use of more impressive DL architectures (particularly transformers) is slowly becoming common in the realms of digital histopathology. However, and recognising that transformers and other multi-resolution/hierarchical approaches can be useful, we tried to reproduce the “hematologist’s approach”: identify regions of the slide which are relevant, detect individual cells and analyze their shape according to understandable and oftentimes simple parameters. We believe that this will appeal to hematologists and hematopathologists. We try to address these points in the Discussion (lines 369-380).

L369-372 “Additionally, neutrophil hyperlobulation was robustly detectable not only in MA but also in IDA55,56 demonstrating that the ability to computationally analyze large numbers of cells can detect this feature even when it is subtle and would otherwise require enumeration of large numbers of neutrophils and their lobe count by experts56.”

- I wonder whether the IDA and MA patients developed any hematological malignancy in the next 5 years? PBS are not routine samples for IDA and MA. Therefore, please confirm that these patients were not associated with hematological malignancy.

We have confirmed that none of these patients developed a hematological malignancy, although many have not been followed up for 5 years yet.

Supplementary methods

L134-136 Considering that this may lead to detecting the same cell twice as the same object, we exclude a prediction if a cell has already been predicted within 8 pixels.

- Why 8 pixels?

This a heuristic value, in essence we want to avoid predicting the same cell twice if it appears near the edge of two tiles (given that we use a sliding window approach). We have added some details to clarify this in the Supplementary Methods (lines 128-130).

L175 “from the MLLS”

- MLL?

Yes, we have now corrected this.

L182 Why 25 epochs? Why not training with callbacks?

While callbacks such as early stopping could have been employed, we observed that 25 epochs was sufficient to guarantee convergence and decided to stop training at this stage.

L183 “every time”

- You mean after every epoch?

Yes, we have now corrected this (line 359).

L181-182 "Tiles were artificially augmented (computationally altered) during training through random flips/rotations/JPEG compression."

L184-186 "During training, each image had a 60% probability of having its brightness, saturation, hue and contrast randomly altered (by 15%, 10%, 10% and 10%, respectively) or of having random JPEG compression artefacts introduced."

- Am I missing something or are the lines 181-182 and 184-186 contradictory to each other?

You are correct, we specified different aspects of the image augmentation at different parts of the text. We have now deleted the first and kept the latter (some values were also slightly off and have now been corrected).

L167 Quality control network

- I would be interested to see if there are any differences in the amount of good vs. poor quality tiles a. per cohort and b. per manually vs. automatically prepared slides.

This is quite an interesting query. We have now assessed this and show that manual preparation is detrimental for the proportion of good quality tiles, while controlling for both cohort and condition (lines 216-224).

L198 You excluded groups of RBCs. Do you think it is possible that these could share biological information that individual RBCs would not?

Reviewer 2 had a similar comment that we have responded to. In essence, we do believe that there is some possible clinical significance in Rouleaux formations (or coinstacks as they are otherwise known) and that they are a feature of specific diseases as noted in ⁸. However, we found it hard to automatically capture these while excluding non-significant clusters of RBC and took the deliberate choice of excluding coinstacks. Importantly, "coinstacks" become more common in thicker parts of the PBS. We have added a comment to the Supplementary Methods to further clarify this (lines 212-218 of the SI).

L257-259 Data augmentation and Supplementary Table S3

- did you manually validate that the augmented data were realistic representations of the true data? This should be confirmed to the supplementary material.

Our objective with data augmentation was not to reproduce likely perturbations to the data, but rather to increase the generalization capabilities of our network. Indeed, according to ¹², the random transformation of histopathology images to yield quite unlikely (or even downright impossible) images leads to better generalization capabilities, outperforming methods which incorporate stain and color normalization (for clarity please refer to Table 1 in ¹²). We have clarified our reasoning behind these augmentations in the Supplementary Methods (lines 260-263).

L266-268 "Prediction post-processing is done by removing objects detected as WBC whose size lies outside the expected distribution for WBC sizes, and filling small convex hull defects."

- How did you define these reference values?

We established the minimum and maximum of acceptable WBC pixel sizes from those available in our training data and rounded down or up (respectively) to the nearest -thousand. We have clarified this in the Supplementary Methods (lines 287-290).

Supplementary Figure S2

- Why is the IoU substantially lower in CUH2 with depth mult. 0.25 than in CUH1 and MLL?

We believe that the U-Net with depth mult. 0.25 is under-parameterized and cannot capture the variation that may be introduced by using different scanners. We have added a comment on this to the Supplementary Results (lines 497-500).

L276-278 To account for different resolutions (0.2268 micrometers/pixel for Hamamatsu NanoZoomer 2.0; 0.2517micrometers/pixel for Aperio AT2) we rescale the images in CUH2 by a factor of 1.1098 prior to cellular characterization.

- I have difficulties to understand this. You might be right in doing this but please explain more why this is needed. Won't this inflict a bias on the cell perimeter?

It is the absence of the rescaling that would introduce the bias on the cell perimeter/area. A caveat we had to deal with when using different scanners was the different resolutions for each; without this rescaling step, we would have, in effect, a pixel representing different biological/clinical units of dimension depending on the scanner. While in the future it could be interesting to have algorithms that are able, much like humans, to generalize beyond pixel resolution, we chose to minimize the effect that this could have by transforming all images to a common resolution.

Supplementary Figure S5

- I find the figure, especially A and C so important that these should be in the main figures.

We confess that this was originally our plan but we feared we may confuse readers by providing an excess of metrics for different models. We have nonetheless added a new figure (Figure 3, made up of previous Sup Fig 5a,c) and renumbered other Figures accordingly (to avoid cluttering the text we have refrained from signaling these with highlights).

- Did you normalize the features for glmnet? If not then, the coefficient (NB! Correct the x-axis label) does not reflect feature importance!

The features are normalized (i.e. standardized, as noted in the now corrected x-axis label). We have now clarified this in the figure legend.

- Age, gender and cytopenias are known to associate with bone marrow morphology (Brück et al Blood Cancer Discovery 2021). Based on Fig 1, the cohorts differed by these factors. Please show, if you can predict age, cytopenias and gender with the features (if feasible, separately in normal, MDS and anaemia patients). If yes, then the authors should exclude these features from the other models to find true age- and gender-independent features.

While this suggestion is relevant for our work, it is hard to perform this for the whole dataset – the demographics of the individuals with MDS in our training dataset is representative of those observed in the real world, implying that more men than women and people of older age have MDS; for this reason, excluding features which are strongly associated with age or gender can lead to the exclusion of features which are also associated with MDS (this is also the main reason why we avoided using age and gender as predictors in our models, as they could artificially absorb much of the predictive signal in our data without contributing towards more accurate explanations of disease). As for performing this analysis separately in normal, MDS and anaemia, this also proves challenging – in effect we have fewer normal and anaemia cases. Nonetheless, we have carried out this analysis for MDS cases, summarised in the Supplementary Results (lines 541-572).

We should, however, note here that simply removing features associated with cytopenias or age/sex may control some specific types of confounding effects (i.e. a cause other than MDS is responsible for causing changes in both WBC counts and morphometry), but it can also dilute the morphometric signal in our data – indeed, if MDS causes both (for the sake of argument) changes in WBCC and morphometry, removing morphometric features which are associated with WBCC will simply remove features which are associated

with MDS and not (directly) with WBC. With no knowledge of the causal structure of our data, we feel this approach would be excessively conservative, and instead chose to reinforce in the manuscript that: i) there is indeed some degree of association between morphometry and cytopenias and ii) we control for these in all models from which we derive conclusions regarding morphometric moments or morphotypes. Additionally, we do not include age or sex as our analysis shows our morphometric features are not associated with either age or sex (Supplementary Results, section “Avoiding demographic and blood count confounders in determining morphometric trends”).

Rebuttal references

1. Fu, Y. *et al.* Pan-cancer computational histopathology reveals mutations, tumor composition and prognosis. *Nat Cancer* **1**, 800–810 (2020).
2. Alomari, Y. M., Sheikh Abdullah, S. N. H., Zaharatul Azma, R. & Omar, K. Automatic detection and quantification of WBCs and RBCs using iterative structured circle detection algorithm. *Comput. Math. Methods Med.* **2014**, 979302 (2014).
3. Elsalamony, H. A. Healthy and unhealthy red blood cell detection in human blood smears using neural networks. *Micron* **83**, 32–41 (2016).
4. Tomari, R., Zakaria, W. N. W., Jamil, M. M. A., Nor, F. M. & Fuad, N. F. N. Computer Aided System for Red Blood Cell Classification in Blood Smear Image. *Procedia Comput. Sci.* **42**, 206–213 (2014).
5. Delgado-Font, W. *et al.* Diagnosis support of sickle cell anemia by classifying red blood cell shape in peripheral blood images. *Med. Biol. Eng. Comput.* **58**, 1265–1284 (2020).
6. Sunarko, B. *et al.* Red blood cell classification on thin blood smear images for malaria diagnosis. *J. Phys. Conf. Ser.* **1444**, 012036 (2020).
7. Chadha, G. K., Srivastava, A., Singh, A., Gupta, R. & Singla, D. An Automated Method for Counting Red Blood Cells using Image Processing. *Procedia Comput. Sci.* **167**, 769–778 (2020).
8. Abramson, N. Rouleaux formation. *Blood* **107**, 4205 (2006).
9. Findlater, R. R. & Schnell-Hoehn, K. N. When blood runs cold: cold agglutinins and cardiac surgery. *Can. J. Cardiovasc. Nurs.* **21**, 30–4; quiz 35–6 (2011).
10. Vardiman, J. W. *et al.* The 2008 revision of the World Health Organization (WHO) classification of myeloid neoplasms and acute leukemia: rationale and important changes. *Blood* **114**, 937–951 (2009).
11. Caicedo, J. C. *et al.* Nucleus segmentation across imaging experiments: the 2018 Data Science Bowl. *Nat. Methods* **16**, 1247–1253 (2019).
12. Tellez, D. *et al.* Quantifying the effects of data augmentation and stain color normalization in convolutional neural networks for computational pathology. *Med. Image Anal.* **58**, 101544 (2019).

Reviewers' Comments:

Reviewer #1:

Remarks to the Author:

I have no additional comments

Reviewer #2:

Remarks to the Author:

The revised manuscript has well addressed the reviewer's comments, and the reviewer does not have further comments but recommends acceptance of the manuscript.

Reviewer #3:

Remarks to the Author:

Revision of "Computational analysis of peripheral blood smears detects disease-associated cytomorphologies" by José Guilherme de Almeida et al.

The authors have addressed the reviewers' comments in detail and have modified the manuscript in response to their suggestions. No additional revision requests.